# Semantic Self-adaptation:
# Enhancing Generalization with a Single Sample

**Sherwin Bahmani**[*1]   **Oliver Hahn**[*1]   **Eduard Zamfir**[*†2]
**Nikita Araslanov**[†3]   **Daniel Cremers**[3]   **Stefan Roth**[1,4]
[1]TU Darmstadt   [2]University of Würzburg   [3]TU Munich   [4]hessian.AI

**Reviewed on OpenReview:** `https://openreview.net/forum?id=ILNqQhGbLx`

## Abstract

The lack of out-of-domain generalization is a critical weakness of deep networks for semantic segmentation. Previous studies relied on the assumption of a static model, *i.e.*, once the training process is complete, model parameters remain fixed at test time. In this work, we challenge this premise with a *self-adaptive* approach for semantic segmentation that adjusts the inference process to each input sample. Self-adaptation operates on two levels. First, it fine-tunes the parameters of convolutional layers to the input image using consistency regularization. Second, in Batch Normalization layers, self-adaptation interpolates between the training and the reference distribution derived from a single test sample. Despite both techniques being well known in the literature, their combination sets new state-of-the-art accuracy on synthetic-to-real generalization benchmarks. Our empirical study suggests that self-adaptation may complement the established practice of model regularization at training time for improving deep network generalization to out-of-domain data. Our code and pre-trained models are available at https://github.com/visinf/self-adaptive.

## 1 Introduction

The current state of the art for semantic segmentation (Chen et al., 2018b; Long et al., 2015) lacks direly in out-of-distribution robustness, *i.e.*, when the training and testing distributions are different. Numerous studies have investigated this issue, primarily focusing on image classification (Arjovsky et al., 2019; Bickel et al., 2009; Li et al., 2017a; Torralba & Efros, 2011; Volpi et al., 2018). However, a recent study of existing domain generalization methods (Gulrajani & Lopez-Paz, 2021) comes to a sobering conclusion: Empirical Risk Minimization (ERM), which makes an *i.i.d.* assumption of the training and testing samples, is still highly competitive. This is in stark contrast to the evident advances in the area of domain adaptation, both for image classification (Ben-David et al., 2010; Ganin et al., 2016; Long et al., 2016; Xie et al., 2018) and semantic segmentation (Araslanov & Roth, 2021; Vu et al., 2019; Yang & Soatto, 2020). This setup, however, assumes access to an unlabelled test distribution at training time. In contrast, in the generalization setting considered here, *only one test sample is accessible at inference time* and *no knowledge between the subsequent test samples must be shared*.

In this work, we study the generalization problem of semantic segmentation from synthetic data (Richter et al., 2016; Ros et al., 2016) through the lens of adaptation. In contrast to previous work that modified the model architecture (Pan et al., 2018) or the training process (Chen et al., 2021; 2020; Yue et al., 2019), we revise the standard *inference* procedure with a technique inspired by domain adaptation methods (Araslanov & Roth, 2021; Li et al., 2017b). The technique, that we term *self-adaptation*, leverages a self-supervised loss, which allows for adapting to a single test sample with a few parameter updates. Complementary to these loss-based updates, self-adaptation integrates feature statistics of the training data with those of the test sample in the Batch Normalization layers (Ioffe & Szegedy, 2015), commonly employed in modern convolutional neural

---

*Equal contribution;   † work primarily done while at TU Darmstadt.

networks (CNNs) (He et al., 2016). Expanding upon the previous conclusions in related studies (Schneider et al., 2020), we find that this normalization strategy not only improves the segmentation accuracy, but also the calibration quality of the prediction confidence.

In summary, our contributions are the following: *(i)* We propose a self-adaptive process for model generalization in the context of semantic segmentation, where model inference adjusts to each test sample. *(ii)* We overcome deficiencies in the experimental protocol used in previous studies with a rigorous revision. *(iii)* Implemented on top of a simple baseline, self-adaptation surpasses previous work and achieves new state-of-the-art segmentation accuracy in synthetic-to-real generalization.

## 2   Related work

Our work contributes to recent research on generalization of semantic segmentation models, and relates to studies on feature normalization (Pan et al., 2018; Schneider et al., 2020) and online learning (Sun et al., 2020). While the focus in previous investigations was the training strategy (Yue et al., 2019) and model design (Pan et al., 2018), we exclusively study the test-time inference process here. Yue et al. (2019) augment the synthetic training data by transferring style from real images. Assuming access to a classification model trained on real images, Chen et al. (2020) regularize the training on synthetic data by ensuring feature proximity of the two models via distillation, and seek layer-specific learning rates for improved generalization. Advancing the distillation technique, Chen et al. (2021) devise a contrastive loss that facilitates model invariance to standard image augmentations. Pan et al. (2018) heuristically add instance normalization (IN) layers to the network. More recently, Choi et al. (2021) and Huang et al. (2021) extract domain-invariant feature statistics by either using an instance-selective whitening loss or frequency-based domain randomization. Similarly, Nam et al. (2022) learn a style-invariant representation by using a causal framework for data generation. Kundu et al. (2021) increase source domain diversity by augmenting single-domain data to virtually simulate a multi-source scenario. Tang et al. (2021) swap channel-wise statistics in feature normalization layers and learn adapter functions to re-adjust the mean and variance based on the input sample. Lee et al. (2022a) enforces consistency of the output logits across multiple images (or pixels) of the same class. To improve generalization in federated learning, Caldarola et al. (2022) train clients locally with sharpness-aware minimization and averaging stochastic weights. However, these methods assume access to a *distribution* of real images during training (Chen et al., 2020; 2021; Yue et al., 2019) (as opposed to only for pre-training of the backbone), or require a modification of the network architecture (Pan et al., 2018). Our work requires neither, hence the presented technique applies even *post-hoc* to the already (pre-)trained models to improve their generalization. Moreover, as we discuss in Sec. 5, the evaluation protocol used by previous studies exhibits a number of shortcomings, which we also address.

**Normalization.**   Batch Normalization (BN; Ioffe & Szegedy, 2015) and other normalization techniques have been increasingly linked to model robustness (Deecke et al., 2019; Huang et al., 2019; Schneider et al., 2020; Wang et al., 2018; Wu & He, 2018). The most commonly used BN, Layer Normalization (LN; Ba et al., 2016), and Instance Normalization (IN; Ulyanov et al., 2016) also affect the model's expressive power, which can be further enhanced by combining these techniques in one architecture (Luo et al., 2019; Nam & Kim, 2018). In a domain adaptation setting, Li et al. (2017b) use source-domain statistics during training while replacing them with target-domain statistics during inference. More recently, Schneider et al. (2020) combine the source and target statistics during inference, but the statistics are weighted depending on the number of samples that these statistics aggregate. Nado et al. (2020) propose using batch statistics during inference from the target domain instead of the training statistics acquired from the source domain. Our comprehensive empirical study complements these results by demonstrating improved generalization of semantic segmentation models.

**Adapting the model to a test sample.**   Several examples from previous work update the model parameters at the time of inference. In object tracking, the object detector has the need to adjust to the changing appearance model of the tracked instance (Kalal et al., 2012). Conditional generative models can learn from a single image sample for the task of super-resolution (Glasner et al., 2009) and scene synthesis (Shaham et al., 2019). More recently, this principle has been used for improving the robustness of image classification models (Sun et al., 2020; Wang et al., 2021a; Zhang et al., 2022). The design of the self-supervised task to perform on the test sample is crucial, and the techniques developed for image classification do not always

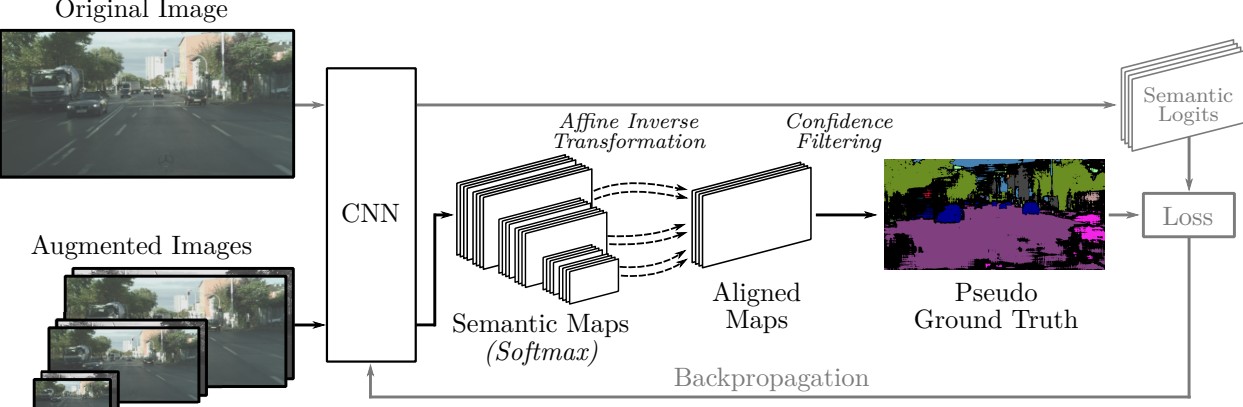

Figure 1: *Overview of the one-sample adaptation process.* We augment a single test sample by creating a batch of images at multiple scales, each with horizontal flipping and grayscaling. To transform the output of each version back to the original image plane, we apply the corresponding inverse affine transformation to every prediction. After averaging the softmax probabilities, we create a pseudo-label using a class-dependent confidence threshold. We update the model parameters by minimizing the cross-entropy loss with respect to the pseudo-label, repeating this process for a small number of iterations ($N_t$) before producing the final prediction. The updated model is then discarded.

extend to dense prediction tasks, such as semantic segmentation considered here. Nevertheless, more suitable alternatives for the self-supervised loss have been recently proposed for domain adaptation (Araslanov & Roth, 2021), and a number of other works devised domain-specific approaches for medical imaging (Varsavsky et al., 2020) or first-person vision (Cai et al., 2020). Concurrently to our work, Reddy et al. (2022) enforce edge consistency during inference for unsupervised test-time adaptation of semantic segmentation methods.

**Setup comparison.** Most of these technically related works (Schneider et al., 2020; Sun et al., 2020; Wang et al., 2021a) focus on the problem of domain adaptation in the context of image classification. They typically assume access to a number of samples (or even all test images) from the target distribution at training time. Our work instead addresses semantic segmentation in the domain generalization setting, which is different as it only necessitates *a single datum* from the test set. In this scenario, simple objectives, such as entropy minimization employed by Tent (Wang et al., 2021a), improve the baseline accuracy only moderately. By contrast, our self-adaptation with pseudo-labels accounts for the inherent uncertainty in the predictions, which proves substantially more effective, as the comparison to Tent in Sec. 5.3 reveals. Our task is also different from few-shot learning (*e. g.*, Finn et al., 2017), where the model may adapt at test time using a small *annotated* set of image samples. No such annotation is available in our setup; our model adjusts to the test sample in a completely unsupervised fashion, requires neither proxy tasks to update the parameters (Sun et al., 2020) nor any prior knowledge of the test distribution.

## 3 Self-adaptive learning from a single sample

The traditional assumption at inference time is that the parameters of the segmentation model remain fixed. However, (self-)adaptive systems and their biological counterparts provide an example, where they learn to specialize on the particularities of their environment (Thrun, 1998; Widmer & Kubat, 1996). By analogy, we here allow the segmentation model to update its parameters. Note that our setup is distinct from the domain adaptation scenario (*e. g.*, in contrast to Wang et al., 2021a), since we discard the updated parameters when processing the next sample in line with the underlying concept of domain generalization.

Our approach, visualized in Fig. 1 and summarized in Algorithm 1, uses data augmentation as a method to create mini-batches of images for each test sample. Based on the original test image, we first create a set of $N$ augmented images by multi-scaling, horizontal flipping, and grayscaling. These augmented images are used to form a mini-batch, which is fed through the CNN. We transform the produced softmax probabilities

---

**Algorithm 1:** Summary of self-adaptation.

---

**1** Train segmentation model on source data (best-practice, established methodology)
**2** Replace BatchNorm with SaN (*cf.* Sec. 3.1)
**3** Tune hyperparameter $\alpha$ in SaN on validation set (WildDash)
**4** # Inference on any dataset. Initial model parameters: $\theta_0$.
**5 foreach** *test sample* **do**
**6**   Obtain $\theta^*$ by minimizing cross-entropy w.r.t. pseudo-labels in Eq. (4).
**7**   Predict segmentation for the test sample using $\theta^*$.
**8**   Reset model parameters to $\theta_0$.
**9 end**

---

from the model back to the original pixels using the inverse affine transformations, and denote the result as $m_{i,:,:,:}$ for every sample $i$ in the mini-batch. This allows the model to have multiple predictions for one pixel. We then compute the mean $\bar{m}$ of these softmax probabilities along the mini-batch dimension $i$ for class $c$ and pixel $(j,k)$ on the spatial grid as

$$\bar{m}_{c,j,k} = \frac{1}{N}\sum_{i=1}^{N} m_{i,c,j,k}.\tag{1}$$

Using hyperparameter $\psi \in (0,1)$, we compute a threshold value $t_c$ from the maximum probability of every class to yield a class-dependent threshold $t_c$:

$$t_c = \psi \cdot \max(\bar{m}_{c,:,:}).\tag{2}$$

Finally, for every pixel, we extract the class $c^*_{j,k}$ with the highest probability by

$$c^*_{j,k} = \arg\max(\bar{m}_{:,j,k}).\tag{3}$$

We ignore low-confidence predictions using our class-dependent threshold $t_c$. Specifically, all pixels with a softmax probability below the threshold are set to an ignore label, while the remaining pixels use the dominant class $c^*_{j,k}$ as the pseudo-label $u_{j,k}$,

$$u_{j,k} = \begin{cases} c^*_{j,k}, & \text{if } \max(\bar{m}_{:,j,k}) \geq t_{c^*_{j,k}} \\ \text{ignore}, & \text{otherwise.} \end{cases}\tag{4}$$

The pseudo ground truth $u$ for the test image is used to fine-tune the model for $N_t$ iterations with gradient descent using the cross-entropy loss. We determine all hyperparameters, *i. e.*, resolution of the scales, threshold $\psi$, number of iterations $N_t$, and learning rate $\eta$, based on a validation dataset. After the self-adaptation process, we produce a single final prediction using the updated model weights. To process the next test sample, we reset these weights to their initial value, hence the model obtains no knowledge about the complete target data distribution.

One natural consideration in this process is the quality of the confidence calibration in the semantic maps, since the $\arg\max$ operation in Eq. (3) and applying the threshold in Eq. (4) aim to select only the most confident pixel predictions. If the confidence values become miscalibrated (*e. g.*, due to the domain shift), a significant fraction of incorrect pixel labels will end up in the pseudo-mask. Our self-adaptive normalization, which we detail next, mitigates this issue.

### 3.1 Self-adaptive normalization

**Batch Normalization** (BN; Ioffe & Szegedy, 2015) has become an inextricable component of modern CNNs (He et al., 2016). Although BN was originally designed for improving training convergence, there is now substantial evidence that it plays an important role in model robustness (Nado et al., 2020), including domain generalization (Pan et al., 2018). Let $x \in \mathbb{R}^{B,H,W}$ denote a spatial tensor of size $H \times W$ for an arbitrary feature channel and batch size $B$, produced by a convolutional layer. We omit the layer and channel indexing,

as the following presentation applies to all layers and feature channels on which BN operates. At training time, BN first computes the mean and the standard deviation across the batch and spatial dimensions, *i. e.*,

$$\mu = \frac{1}{BHW} \sum_{i,j,k} x_{i,j,k} \,, \qquad \sigma^2 = \frac{1}{BHW} \sum_{i,j,k} (x_{i,j,k} - \mu)^2. \tag{5}$$

The normalized features $\hat{x}$ follow from applying these statistics:

$$\bar{x}_{i,j,k} = \frac{(x_{i,j,k} - \mu)}{\sqrt{\sigma^2 + \epsilon}}. \tag{6}$$

Notably, this process differs from the normalization used at inference time. At training time, every BN layer maintains a running estimate of $\mu$ and $\sigma$ across the training batches, which we denote here as $\hat{\mu}$ and $\hat{\sigma}$. At test time, it is an established practice to normalize the feature values with $\hat{\mu}$ and $\hat{\sigma}$ in Eq. (6), instead of the test-batch statistics. We refer to this scheme as *train BN* (*t*-BN).

**BN and generalization.** In the context of out-of-distribution generalization, the running statistics $\hat{\mu}$ and $\hat{\sigma}$ derive from the source data and can be substantially different had they been computed using the target images. This discrepancy is generally known as the *covariate shift* problem. Domain adaptation methods, which assume access to the (unlabelled) target distribution, often alleviate this issue with a technique referred to as Adaptive Batch Normalization (AdaBN; Li et al., 2017b). The key idea behind the method is to simply replace the source running statistics with those of the target. Instead of alternating BN layers between the training and testing modes, recent work (Nado et al., 2020) studies *prediction-time BN* (*p*-BN), which replaces the running statistics from the training time, $\hat{\mu}$ and $\hat{\sigma}$, with the statistics $\mu$ and $\sigma$ of the current test batch (normally comprising more than one image). Such a seemingly innocuous change is shown to benefit model robustness for image classification (Nado et al., 2020; Schneider et al., 2020).

In contrast to AdaBN and *p*-BN, which utilize either the whole target distribution or a number of samples, our study of model generalization assumes only a single target example to be available — the one that our model receives as the input at inference time. A viable alternative to AdaBN and *p*-BN is to compute the statistics per sample, which amounts to replacing BN layers with Instance Normalization (IN) layers (Ulyanov et al., 2016) after model training. However, this may cause another extreme scenario for covariate shift. Firstly, the sample statistics may serve only as an approximation to the complete test distribution. Secondly, such a replacement may significantly interfere with the statistics of the activations in the intermediate layers with which the network was trained. It is, therefore, unsurprising that IN layers alone were found to only hurt the discriminative power of the model in previous work (Pan et al., 2018).

**Self-adaptive normalization (SaN).** Let the *source* mean $\hat{\mu}_s$ and the variance $\hat{\sigma}_s^2$ denote the running average of the sample statistics at training time. If we had the *target* domain knowledge expressed by the sufficient statistics $\hat{\mu}_t$ and $\hat{\sigma}_t^2$, we could use those in place of $\hat{\mu}_s$ and $\hat{\sigma}_s^2$ in the BN layers to compensate for the covariate shift. However, at test time we only have access to the sample estimates, $\mu_t$ and $\sigma_t^2$, provided by a single datum from the target distribution:

$$\mu_t = \frac{1}{HW} \sum_{j,k} z_{j,k} \,, \qquad \sigma_t^2 = \frac{1}{HW} \sum_{j,k} \left( z_{j,k} - \mu_t \right)^2, \tag{7}$$

where $z \in \mathbb{R}^{H,W}$ is a spatial feature channel in a CNN of the target sample. We define $\alpha \in [0,1]$, which denotes the change from the source ($\alpha = 0$) to a *reference* ($\alpha = 1$) image domain. At inference time we compute the new mean and variance, $\hat{\mu}_t$ and $\hat{\sigma}_t$, as follows:

$$\hat{\mu}_t := (1-\alpha)\hat{\mu}_s + \alpha\mu_t \,, \qquad \hat{\sigma}_t^2 := (1-\alpha)\hat{\sigma}_s^2 + \alpha\sigma_t^2. \tag{8}$$

Using Eq. (6), we replace $\mu$ and $\sigma^2$ with the computed $\hat{\mu}_t$ and $\hat{\sigma}_t^2$ to normalize the features of the single test sample. Notably, this does not affect the behavior of the BN layers at training time and applies only at test time. Since this approach combines the inductive bias coming in the form of the running statistics from the source domain with statistics extracted from a single test instance, which is an unsupervised process, we refer to it as *Self-adaptive Normalization* (SaN).

In Sec. 5.1, we empirically verify that SaN consistently boosts the segmentation accuracy in the out-of-distribution scenario. Furthermore, we also find significant improvements of model calibration in terms of the expected calibration error (ECE; Naeini et al., 2015).

**Related methods.** Setting $\alpha = 0$ in Eq. (8) defaults to the established procedure, $t$-BN, and uses only the running statistics from training on the source domain at inference time. Conversely, $\alpha = 1$ corresponds to Instance Normalization (Ulyanov et al., 2016) or, equivalently, to the $p$-BN strategy (Nado et al., 2020) with a batch size of 1. While $\mu_t$ and $\sigma_t$ are averaged over a *batch* of target images in (Nado et al., 2020), we rely on a single test sample in this work. Our experiments in Sec. 5.1 expand upon previous analyses (Schneider et al., 2020) with an extensive emperical study of this normalization strategy for semantic segmentation. Batch Instance Normalization (Nam & Kim, 2018) used a similar weighting approach for adaptive stylization. However, $\alpha$ was a *training* parameter, whereas we use $\alpha$ only for model selection using a validation set.

## 4 Designing a principled evaluation

Previous studies (Chen et al., 2020; 2021; Pan et al., 2018; Yue et al., 2019) on domain generalization for semantic segmentation used divergent evaluation methodologies, which exacerbates the comparison and reproducibility in follow-up research. Encouraged by similar observations (Gulrajani & Lopez-Paz, 2021), we first set out and then follow a number of principles reflecting the best-practice experimentation to enable a fair and reproducible evaluation.

*(A) The test set must comprise multiple domains.* – A few previous works (Chen et al., 2020; 2021; Pan et al., 2018) used only a single target domain, Cityscapes (Cordts et al., 2016), for testing. However, like other research datasets, Cityscapes (Cordts et al., 2016) is carefully curated (*e. g.*, the same camera hardware and country was used for image capture), hence only partially represents the visual diversity of the world. A more comprehensive approach by (Yue et al., 2019) considered a number of target domains. However, a separate model was selected for every target domain (based on a *different* validation set). As a result, each selected model may be biased toward the chosen target domain, hence may not be indicative of strong generalization. This leads us to the next principle:

*(B) A single model must be used for all test domains.* – This principle follows naturally from the definition of generalization, *i. e.* the ability of the model to reliably operate (*e. g.* in terms of accuracy) under varying test environments. To produce a domain-specific model is the goal of another research avenue, domain adaptation.

Many works do not clearly define the strategy of model selection, such as the used validation set. Indeed, the use of test domains for hyperparameter tuning has not been uncommon (*e. g.* (Huang et al., 2021)). This is especially undesirable when studying model generalization. Therefore, we stress that:

*(C) Model selection must not use test images;* and *(D) The validation set must be clearly specified.* – These principles follow naturally from the requirements of domain generalization, with principles $C$ and $D$, in particular, being widely accepted in machine learning research. To our surprise, we found that *no previous work on domain generalization for semantic segmentation has yet fulfilled all of these principles.* To encourage and facilitate good evaluation practice, we therefore revise the experimental protocol used so far. We consider a practical scenario in which a supplier prepares a model for a consumer without *a-priori* knowledge on where this model may be deployed, much akin to Kundu et al. (2021). On the supplier's side, we assume access to two data distributions for model training and validation, the *source data* and the *validation set*. After training the model on the source data and choosing its hyperparameters on the validation set, we assess its generalization ability on three qualitatively distinct *target sets*. The average accuracy across these sets provides an estimate of the expected model accuracy for its out-of-distribution deployment on the consumer's side. Next, we concretize the datasets used in this study, which focus on traffic scenes for compatibility with previous work (Chen et al., 2021; Pan et al., 2018; Yue et al., 2019). Notably, the scale of our study, summarised in Table 1, exceeds that of previous work.

In contrast, our evaluation methodology adheres to the principles of robustness and generalization by testing a single model on multiple target domains using a single set of hyperparameters without access to images from respective target datasets. This ensures that our model can perform well across a range of domains without the need for domain-specific fine-tuning or model selection.

Table 1: *State-of-the-art domain generalization methods for reported source domains and target domains.*

| Method | *Source Domains* | | *Target Domains* | | | |
|---|---|---|---|---|---|---|
| | GTA | SYNTHIA | CS | Mapillary | BDD | IDD |
| DRPC (Yue et al., 2019) | ✓ | ✓ | ✓ | ✓ | ✓ | - |
| ASG (Chen et al., 2020) | ✓ | - | ✓ | - | - | - |
| CSG (Chen et al., 2021) | ✓ | - | ✓ | - | - | - |
| RobustNet (Choi et al., 2021) | ✓ | - | ✓ | ✓ | ✓ | - |
| FSDR (Huang et al., 2021) | ✓ | ✓ | ✓ | ✓ | ✓ | - |
| WildNet (Lee et al., 2022b) | ✓ | - | ✓ | ✓ | ✓ | - |
| SAN-SAW (Peng et al., 2022) | ✓ | ✓ | ✓ | ✓ | ✓ | - |
| PIN (Kim et al., 2022) | ✓ | ✓ | ✓ | ✓ | ✓ | - |
| GCISG (Nam et al., 2022) | ✓ | - | ✓ | - | - | - |
| XDED (Lee et al., 2022a) | ✓ | - | ✓ | ✓ | ✓ | - |
| *This work* | ✓ | ✓ | ✓ | ✓ | ✓ | ✓ |

## 5 Experiments

Following our discussion above, we revise the evaluation protocol as follows. On the supplier's side, we assume access to two data distributions for model training and validation, the *source data* and the *validation set*. We assess the generalization ability of the model yielded by the validation process on three qualitatively distinct *target sets*. The average accuracy across these sets provides an estimate of the expected model accuracy for its out-of-distribution deployment on the consumer's side. Next, we concretize the datasets used in this study, which are restricted to traffic scenes for compatibility with previous work (Chen et al., 2021; Pan et al., 2018; Yue et al., 2019) (see supplemental material for dataset details).

**Source data.** We train our model on the training split of two synthetic datasets (mutually exclusive) with low-cost ground truth annotation: GTA (Richter et al., 2016) and SYNTHIA (Ros et al., 2016). Importantly, these datasets exhibit visual discrepancy (*i. e.*, domain shift) *w. r. t.* the real imagery, to which our model needs to generalize.

**Validation set.** For model selection and hyperparameter tuning, we use the validation set of WildDash (Zendel et al., 2018). In our scenario, the validation set is understood to be of limited quantity, owing to its more costly annotation compared to the source data. In contrast to the training set, however, it bears closer visual resemblance to the potential target domains.

**Multi-target evaluation.** Following model selection, we evaluate the single model on three target domains comprising the validation sets from Cityscapes (Cordts et al., 2016), BDD (Yu et al., 2020), and IDD (Varma et al., 2019). The choice of these test domains stems from a number of considerations, such as the geographic origin of the scenes (Cityscapes, BDD, and IDD were collected in Germany, North America, and India, respectively). Geographic distinction as well as substantial differences in data acquisition (*e. g.*, camera properties) of these datasets bring together an assortment of challenges for the segmentation model at test time. Since the deployment site of our model is unknown, we assume a uniform prior over the target domains as our test distribution. Under this assumption, the average of the mean accuracy across our target domains estimates the expected model accuracy.

To compare to previous works, we also evaluate on Mapillary (Neuhold et al., 2017). Mapillary does not publicly disclose the geographic origins of individual samples, hence is unsuitable to identify a potential location bias acquired by the model from the training data. This is possible in our proposed evaluation protocol, since the geographic locations from Cityscapes, BDD, and IDD do not overlap.

**Implementation details.** We implement our framework in PyTorch (Paszke et al., 2019). Our code and pre-trained models are publicly available. We also discuss trivial to implement, but crucially useful training details of our baseline in-depth here and in Appendix A.

Table 2: *(a) Segmentation accuracy using SaN.* We report the mean IoU (%, ↑) on three target domains (Cityscapes, BDD, IDD) across both backbones. *t*-BN denotes train BN (Ioffe & Szegedy, 2015), while *p*-BN refers to prediction-time BN (Nado et al., 2020). *(b) ECE (%, ↓) for SaN and MC-Dropout (Gal & Ghahramani, 2016).* We trained the networks on GTA and report scores for the three target domains (*cf.* Appendix B.1 for results with SYNTHIA).

| Method | (a) *IoU (%, ↑)* | | | | Method | (b) *ECE (%, ↓)* | | | |
|---|---|---|---|---|---|---|---|---|---|
| | CS | BDD | IDD | Mean | | CS | BDD | IDD | Mean |
| ResNet-50 | | | | | ResNet-50 | 37.28 | 35.61 | 27.73 | 33.54 |
| w/ *t*-BN | 30.95 | 28.52 | 32.78 | 30.75 | w/ SaN (*Ours*) | 30.57 | 30.94 | 26.90 | 29.47 |
| w/ *p*-BN | **37.71** | 31.67 | 30.85 | 33.41 | w/ MC-Dropout | 30.29 | 29.80 | 24.17 | 28.09 |
| w/ SaN (*Ours*) | 37.54 | **32.79** | **34.21** | **34.85** | w/ both (*Ours*) | **25.50** | **27.36** | **22.62** | **25.16** |
| ResNet-101 | | | | | ResNet-101 | 35.24 | 33.74 | 27.28 | 32.09 |
| w/ *t*-BN | 32.90 | 32.54 | 30.36 | 31.93 | w/ SaN (*Ours*) | 26.12 | 28.89 | 23.98 | 26.36 |
| w/ *p*-BN | 39.88 | 34.30 | 33.05 | 35.74 | w/ MC-Dropout | 31.30 | 29.95 | 25.15 | 28.80 |
| w/ SaN (*Ours*) | **42.17** | **35.40** | **33.52** | **37.03** | w/ both (*Ours*) | **24.44** | **28.68** | **23.32** | **25.48** |

Following (Pan et al., 2018), our baseline model is DeepLabv1 (Chen et al., 2015) without CRF post-processing, but the reported results also generalize to more advanced architectures (*cf.* Appendix B.3). We use ResNet-50 and ResNet-101 (He et al., 2016) pre-trained on ImageNet (Deng et al., 2009) as backbone. We minimize the cross-entropy loss with an SGD optimizer and a learning rate of 0.005, decayed polynomially with the power set to 0.9. All models are trained on the source domains for 50 epochs with batch size, momentum, and weight decay set to 4, 0.9, and 0.0001, respectively. For data augmentation, we compute crops of random size (0.08 to 1.0) of the original image size, apply a random aspect ratio (3/4 to 4/3) to the crop, and resize the result to $512 \times 512$ pixels. We also use random horizontal flipping, color jitter, random blur, and grayscaling. We train our models with SyncBN (Paszke et al., 2019) on two NVIDIA GeForce RTX 2080 GPUs.

## 5.1 SaN improves segmentation accuracy and prediction uncertainty

For both source domains (GTA, SYNTHIA) in combination with all main target domains (Cityscapes, BDD, IDD), we investigate the influence of $\alpha$ on Self-adaptive Normalization (SaN, Eq. 8) in terms of the IoU. Setting an optimal $\alpha$ for every target domain is infeasible in domain generalization as the target domain during inference is unknown. Instead, we choose the optimal $\alpha$ in steps of 0.1 based on the IoU on the validation set of WildDash. For the ResNet-50 backbone, we attain the highest validation IoU for both training datasets with $\alpha = 0.1$ (see Appendix B.1). Fixing this optimal $\alpha$, we proceed with evaluating our model on the target domains. Table 2 reports the segmentation accuracy with this $\alpha$ that has been optimized on the validation set. In Table 2(a) we report IoU scores for both backbones on generalization from GTA to Cityscapes, BDD, and IDD and compare the accuracy of the target domains with *t*-BN and *p*-BN. Remarkably, SaN improves the mean IoU not only of the *t*-BN baseline (*e. g.*, by 4.1% IoU with ResNet-50), which represents an established evaluation mode, but also over the more recent *p*-BN (Nado et al., 2020). This improvement is consistent across the board, *i. e.* irrespective of the backbone architecture and the target domain tested. Furthermore, we found that the calibration of our models, in terms of the expected calibration error (ECE; Naeini et al., 2015), also improves. As shown in Table 2(b), not only does SaN substantially enhance the baseline, but is even competitive with the commonly used MC-Dropout method (Gal & Ghahramani, 2016). Rather surprisingly, SaN exhibits a complementary effect with MC-Dropout: the calibration of the predictions improves even further when both methods are used jointly.

## 5.2 Self-adaptation *vs.* Test-Time Augmentation (TTA)

We compare our self-adaptation to the standard non-adaptive inference, as well as test our models against Test-Time Augmentation (TTA; Simonyan & Zisserman, 2015) as a stronger baseline. TTA augments the test samples with their flipped and grayscaled version on multiple scales and averages the predictions as the

Table 3: *Mean IoU (%) with TTA (Simonyan & Zisserman, 2015) and our self-adaptation* reported across both source domains (GTA, SYNTHIA) and three target domains (Cityscapes, BDD, IDD).

| Method | Source: GTA | | | | Source: SYNTHIA | | | |
|---|---|---|---|---|---|---|---|---|
| | CS | BDD | IDD | Mean | CS | BDD | IDD | Mean |
| ResNet-50 (w/ SaN) | 37.54 | 32.79 | 34.21 | 34.85 | 36.14 | 26.66 | 26.37 | 29.72 |
|   TTA (w/ SaN) | 42.56 | 37.72 | 37.98 | 39.42 | 39.67 | 32.10 | 30.46 | 34.08 |
|   Self-adaptation *(Ours)* | **45.13** | **39.61** | **40.32** | **41.69** | **41.60** | **33.35** | **31.22** | **35.39** |
| ResNet-101 (w/ SaN) | 42.17 | 35.40 | 33.52 | 37.03 | 38.01 | 28.66 | 27.28 | 31.32 |
|   TTA (w/ SaN) | 44.37 | 38.49 | 38.35 | 40.40 | 39.91 | 32.68 | 30.04 | 34.21 |
|   Self-adaptation *(Ours)* | **46.99** | **40.21** | **40.56** | **42.59** | **42.32** | **33.27** | **31.40** | **35.66** |

final result. For self-adaptation, we use horizontal flipping and grayscaling with factor scales of (0.25, 0.5, 0.75) $w.\,r.\,t.$ the original image resolution. We study the relative importance of these augmentation types in Appendix B.2. Based on the validation set WildDash, we set threshold $\psi = 0.7$, $N_t = 10$ iterations, and a learning rate $\eta = 0.05$. We only train the layers conv4_x, conv5_x, and the classification head as we did not observe any benefits from updating all model parameters. Furthermore, this reduces runtime due to not backpropagating through the whole network. We investigate this choice as part of the runtime-accuracy trade-off in Sec. 5.4. In Table 3, we show IoU scores for both source domains (GTA, SYNTHIA) and three target domains (Cityscapes, BDD, IDD) across both backbones. Even though TTA improves the baseline (*e. g.*, by 3.37% IoU with ResNet-101 using GTA), our proposed self-adaptation still outperforms it by a clear and consistent margin of 2.19% IoU on average. This observation aligns well with our reported ECE scores in Table 2(b) to demonstrate that self-adaptation further exploits the calibrated confidence of our predictions to yield reliable pseudo-labels for adapting the model to a particular test sample.

### 5.3 Comparison to state of the art

We compare self-adaptation with state-of-the-art domain generalization methods in Table 4. While most of the other methods report their results on weakly tuned baselines, we show consistent improvements even over a carefully tuned baseline, regardless of backbone architecture or source data. Our single model with self-adaptation even outperforms DRPC Yue et al. (2019) and FSDR Huang et al. (2021) on most benchmarks (*e. g.*, by $4.2 - 9.4\%$ on Mapillary with ResNet-101). These methods train individual models for each target domain; FSDR Huang et al. (2021) resorts to the target domains for hyperparameter tuning, hence contravenes our proposed out-of-distribution evaluation protocol. Recall that ASG Chen et al. (2020) and CSG Chen et al. (2021) (as well as DRPC; Yue et al., 2019) require access to a distribution of real images for training, while IBN-Net Pan et al. (2018) modifies the model architecture. Our thoroughly straightforward approach requires neither, alters only the inference procedure, yet outperforms these methods in all benchmark scenarios substantially. WildNet Lee et al. (2022b) appears to be more accurate than self-adaptation on BDD with ResNet-101. However, this is not a fair comparison, since it uses a more advanced architecture (DeepLabv3+ *vs.* DeepLabv1). As we will see in Table 12, self-adaptation, in fact, outperforms WildNet when using the same architecture by 2.79% IoU. Similarly, SAN-SAW Peng et al. (2022) reaches higher accuracy on BDD, if trained on SYNTHIA, presumably due to the ASPP module Chen et al. (2018a) that we do not use. Self-adaptation considerably outperforms SAN-SAW in all other scenarios. Overall, despite adhering to a stricter evaluation practice and a simpler model architecture, self-adaptation overwhelmingly exceeds the segmentation accuracy of previous work.

**Comparison to Tent (Wang et al., 2021a).** Like self-adaptation, Tent (Wang et al., 2021a) also updates model parameters at test time. However, different from constructing the pseudo-labels based on well-calibrated predictions in our self-adaptation, Tent simply minimizes the entropy of a single-scale prediction. Tent also limits the adaptation to updating only the BN parameters, whereas our self-adaptation generalizes this process to convolutional layers. To demonstrate these advantages, we compare to Tent (Wang et al., 2021a) in Table 5. We train HRNet-W18 (Wang et al., 2021b) on GTA and compare the IoU on Cityscapes to the

Table 4: *Mean IoU (%, ↑) comparison to state-of-the-art domain generalization methods* for both source domains (GTA, SYNTHIA) as well as three target domains (Cityscapes, Mapillary, BDD). In-domain training to obtain the upper bounds uses our baseline DeepLabv1 following the same schedule as with the synthetic datasets. (‡), (†), (‡‡) and (††) denote the use of FCN (Long et al., 2015), DeepLabv2 (Chen et al., 2018a), DeepLabv3 (Chen et al., 2017), and DeepLabv3+ (Chen et al., 2018b), respectively.

| | Method | Backbone: ResNet-50 | | | Backbone: ResNet-101 | | |
| --- | --- | --- | --- | --- | --- | --- | --- |
| | | CS | Mapillary | BDD | CS | Mapillary | BDD |
| | In-domain Bound | 71.23 | 58.39 | 58.53 | 73.84 | 62.81 | 61.19 |
| GTA | No Adapt / DRPC‡ (Yue et al., 2019) | 32.45 / 37.42 ↑4.97 | 25.66 / 34.12 ↑8.46 | 26.73 / 32.14 ↑5.41 | 33.56 / 42.53 ↑8.97 | 28.33 / 38.05 ↑9.72 | 27.76 / 38.72 ↑10.96 |
| | No Adapt / ASG† (Chen et al., 2020) | 25.88 / 29.65 ↑3.77 | – | – | 29.63 / 32.79 ↑3.16 | – | – |
| | No Adapt / CSG† (Chen et al., 2021) | 25.88 / 35.27 ↑9.39 | – | – | 29.63 / 38.88 ↑9.25 | – | – |
| | No Adapt / RobustNet†† (Choi et al., 2021) | 28.95 / 36.58 ↑7.63 | 28.18 / 40.33 ↑12.15 | 25.14 / 35.20 ↑10.06 | | – | – |
| | No Adapt / FSDR‡ (Huang et al., 2021) | – | – | – | 33.4 / 44.8 ↑11.4 | 27.9 / 43.4 ↑15.5 | 27.3 / 41.20 ↑13.9 |
| | No Adapt / WildNet†† (Lee et al., 2022b) | 35.16 / 44.62 ↑9.46 | 31.29 / 46.09 ↑14.77 | 29.71 / 38.42 ↑8.71 | 35.73 / 45.79 ↑10.06 | 33.42 / 47.08 ↑13.66 | 34.06 / **41.73** ↑7.67 |
| | No Adapt / SAN-SAW (Peng et al., 2022) | 29.32 / 39.75 ↑10.43 | 28.33 / 41.86 ↑13.53 | 25.71 / 37.34 ↑11.63 | 30.64 / 45.33 ↑14.69 | 28.65 / 40.77 ↑12.12 | 27.82 / 41.18 ↑13.36 |
| | No Adapt / PIN (Kim et al., 2022) | 31.60 / 41.00†† ↑9.40 | 29.00 / 37.40†† ↑8.40 | 25.10 / 34.60†† ↑9.50 | – / 44.90† | – / 39.71† | – / 41.31† |
| | No Adapt / GCISG‡‡ (Nam et al., 2022) | 26.67 / 39.01 ↑12.34 | – | – | – | – | – |
| | No Adapt / XDED†† (Lee et al., 2022a) | 28.90 / 39.20 ↑10.30 | 28.10 / 37.10 ↑9.00 | 25.10 / 32.40 ↑7.30 | – | – | – |
| | No Adapt / Self-adaptation *(Ours)* | 30.95 / **45.13** ↑14.18 | 34.56 / **47.49** ↑12.93 | 28.52 / **39.61** ↑11.09 | 32.90 / **46.99** ↑14.09 | 36.00 / **47.49** ↑11.49 | 32.54 / 40.21 ↑7.67 |
| SYNTHIA | No Adapt / DRPC‡ (Yue et al., 2019) | 28.36 / 35.65 ↑7.29 | 27.24 / 32.74 ↑5.50 | 25.16 / 31.53 ↑6.37 | 29.67 / 37.58 ↑7.91 | 28.73 / 34.12 ↑5.39 | 25.64 / 34.34 ↑8.70 |
| | No Adapt / FSDR‡ (Huang et al., 2021) | – | – | – | - / 40.8 | - / 39.6 | - / **37.4** |
| | No Adapt / SAN-SAW (Peng et al., 2022) | 23.18 / 38.92 ↑15.74 | 21.79 / 34.52 ↑12.73 | 24.50 / **35.24** ↑10.74 | 23.85 / 40.87 ↑17.02 | 21.84 / 37.26 ↑15.42 | 25.01 / 35.98 ↑10.97 |
| | No Adapt / Self-adaptation *(Ours)* | 31.83 / **41.60** ↑9.77 | 33.41 / **41.21** ↑7.80 | 24.30 / 33.35 ↑9.05 | 37.25 / **42.32** ↑5.07 | 36.84 / **41.20** ↑4.36 | 29.32 / 33.27 ↑3.95 |

Table 5: *Mean IoU (%, ↑) comparison to Tent (Wang et al., 2021a). (a)* Using the same HRNet-W18 backbone (Wang et al., 2021b) as in (Wang et al., 2021a), our self-adaptation outperforms Tent (Wang et al., 2021a) substantially, even when using SaN alone. *(b)* Our self-adaptation loss defined in Sec. 3 yields significantly higher segmentation accuracy compared to the entropy minimization used by Tent (Wang et al., 2021a).

(a) HRNet-W18 backbone

| Method | CS |
| --- | --- |
| Tent (Wang et al., 2021a) | 36.4 |
| SaN *(Ours)* | 40.0 |
| Self-adaptation *(Ours)* | **44.1** |

(b) Comparison to entropy minimization (DeepLabv1)

| Method | CS | BDD | IDD | Mean |
| --- | --- | --- | --- | --- |
| Entropy min. | 40.20 | 35.40 | 33.97 | 36.52 |
| *Ours* ($\psi=0.0$) | 44.00 | 38.55 | 39.09 | 40.55 |
| *Ours* ($\psi=0.7$) | **45.13** | **39.61** | **40.32** | **41.69** |

equivalent configuration of Tent. Under a comparable computational budget of 10 model update iterations, we substantially outperform Tent, by a remarkable 7.5% IoU. SaN alone already outperforms Tent significantly with a single forward pass by 3.6%. Further, our self-adaptation loss is also considerably more effective

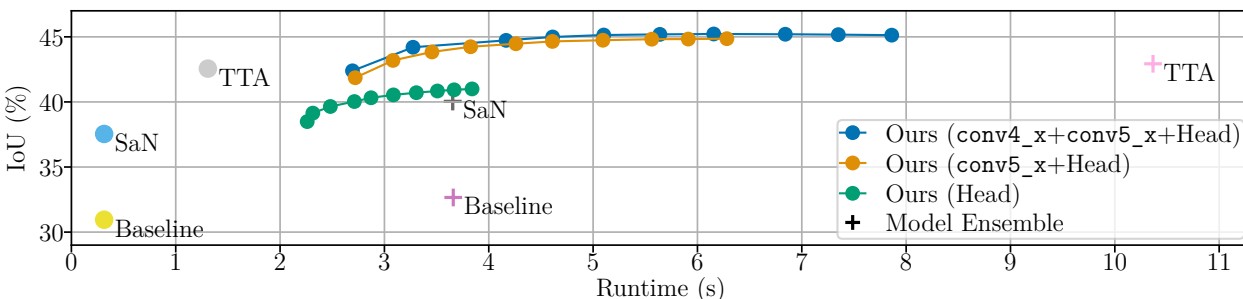

Figure 2: *Runtime-accuracy comparison on GTA → Cityscapes generalization using one NVIDIA GeForce RTX 2080 GPU.* The curves trace self-adaptation iterations, *i.e.*, the first point corresponds to $N_t = 1$, while the last shows $N_t = 10$. Self-adaptation balances accuracy and inference time by adjusting iteration numbers and layer choices, and is more cost-effective than 10 network ensembles.

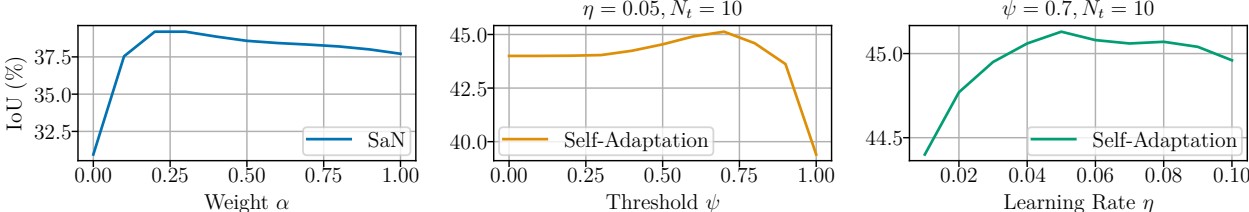

Figure 3: *Hyperparameter sensitivity on GTA → Cityscapes generalization.* We investigate our hyperparameters $\alpha$, threshold $\psi$ and learning rate $\eta$ and report scores using Deeplabv1 with the ResNet-50 backbone trained on GTA. For self-adaptation, we fix one hyperparameter $(\psi, \eta)$ while varying the other.

than the entropy minimization employed by Tent. We improve by 4% IoU on average even without tuning threshold $\psi$, and by 5.2% when it is tuned.

## 5.4 Additional analysis

**Runtime-accuracy trade-off.** We investigate the influence of the number of iterations required to adapt to a single sample during self-adaptation. Fig. 2 plots IoU scores for Cityscapes using the ResNet-50 backbone trained on GTA (Appendix B.2 provides a numerical comparison). As a widely adopted baseline, we also train a model ensemble comprising 10 networks with a DeepLabv1 architecture (as in self-adaptation), initialized with a random seed (Hansen & Salamon, 1990). Note that TTA, self-adaptation, and the ensemble use SaN for a fair comparison, and we also test the ensemble with TTA. Although self-adaptation increases the accuracy of the baselines at the expense of test-time latency, it is still more efficient and more accurate than the model ensembles. Another advantage is that self-adaptation can trade off the accuracy *vs.* runtime by using fewer update iterations, or updating fewer upper network layers. While the top-accuracy variant of self-adaptation may not be suitable for real-time applications yet, it can still be valuable in other important domains, such as medical imaging, where high accuracy is desirable even at the cost of increased latency. For real-time needs, SaN alone boosts the baseline accuracy significantly without any computational overhead.

**Hyperparameter sensitivity** Fig. 3 plots segmentation accuracy *w.r.t.* our hyperparameters $\alpha$, $\psi$, and $\eta$. Complying with our protocol in Sec. 4, we exclusively rely on our validation set to find the optimal values as follows. As detailed in Sec. 5.1, we first use SaN alone and determine $\alpha$ leading to the highest accuracy on the validation set, yielding $\alpha = 0.1$. Recall that the threshold $\psi$ plays a crucial role in determining the reliability of our pseudo-labels, while the learning rate $\eta$ governs the test-time adaptation process. We use grid search and fine-tune network parameters with self-adaptation on the validation set, which establishes the optimal values of $\psi = 0.7$ and $\eta = 0.05$. We observe from Fig. 3 that the model accuracy on the validation set declines moderately as we deviate from the optimal hyperparameter values. For example, if we were to increase $\eta$ twofold, the accuracy would drop by only 0.15% IoU. Appendix B.2 provides numerical results to reproduce

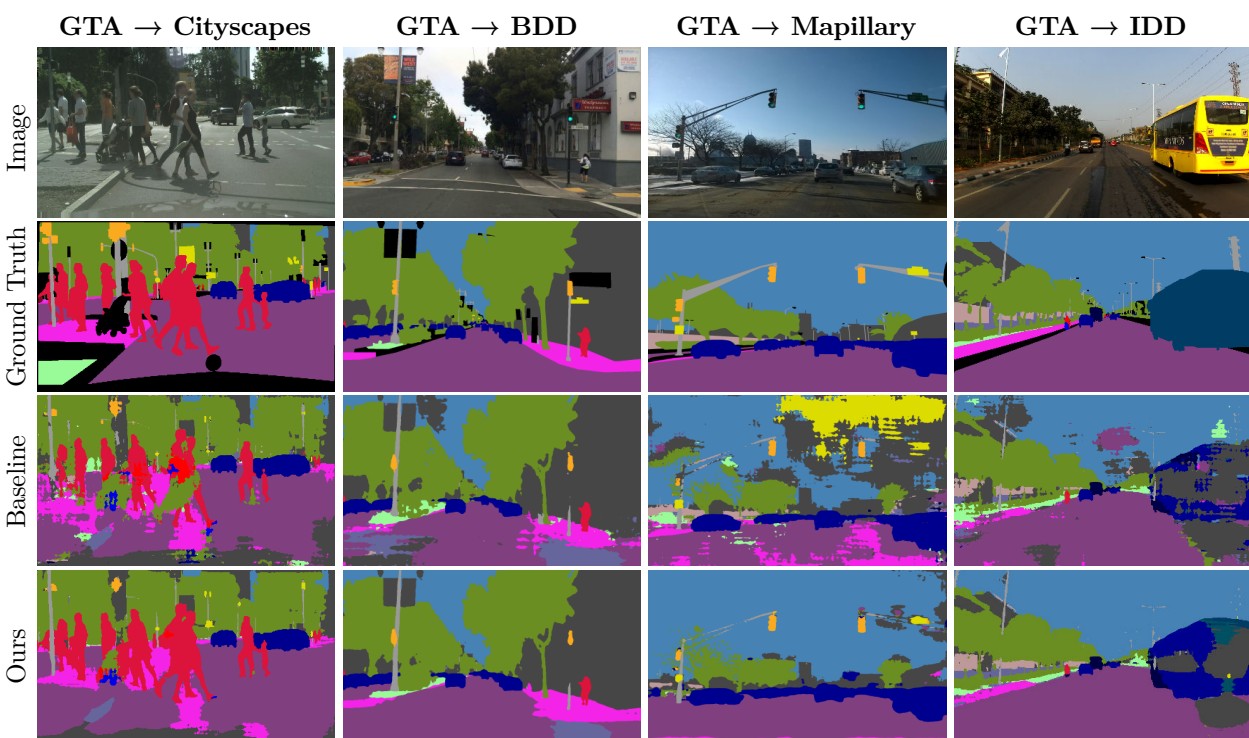

Figure 4: *Qualitative semantic segmentation results for the generalization from GTA to Cityscapes, BDD, Mapillary, and IDD* for the ResNet-50 backbone. We show the input image (top row), ground truth and the predictions of the baseline model and of our proposed self-adaptation (bottom row).

Fig. 3. Overall, hyperparameters in self-adaptation exhibit a reasonable range of tolerance. Our comparision to the state of the art in Sec. 5.3 also supports this conclusion, since these hyperparameter values may not be optimal for the target domains. Self-adaptation achieves superior segmentation accuracy nonetheless.

**Qualitative results.** In Fig. 4 we visualize qualitative segmentation results produced by self-adaptation for generalization from GTA to Cityscapes, BDD, Mapillary, and IDD. We observe a clearly perceivable improvement over the baseline, especially in terms of consistency with the image boundaries. Appendix B.5 illustrates further examples and discusses failure cases.

## 6 Conclusion

The *i.i.d.* assumption underlying the traditional learning principle, ERM, implies that a training process relying on it is unlikely to produce models robust to an arbitrary domain shift, unless we make further assumptions about the test distribution. In the out-of-distribution scenario, the test domain is unknown, hence formulating such assumptions is difficult. To bypass this issue, we presented and studied a self-adaptive *inference* process. We also highlighted a number of shortcomings in the experimental protocol used in previous work. By following the best practice in machine learning research, we formulated four principles defining a rigorous evaluation process in domain generalization. We implemented and followed these principles in our experiments.

Our analysis demonstrates that a single sample from the test domain can already suffice to improve model predictions. The accuracy improvement shown by our experiments is surprisingly substantial, despite using a fairly straightforward approach without changes to the training process or the model architecture, unlike in previous works (Chen et al., 2020; Yue et al., 2019). We hope that these encouraging results will incentivize our research community to study self-adaptive techniques in other application domains, such as panoptic segmentation, or monocular depth prediction.

The particular instantiation of self-adaptation that we presented in this work is not yet real-time. We extensively analyzed the existing trade-off with the segmentation accuracy in Sec. 5.4 and Appendix B.2, and found self-adaptation to be nonetheless more cost-effective than model ensembles — a widely adopted framework in machine learning. However, the scope of this work was to rigorously demonstrate concrete accuracy benefits, which will make improving model efficiency a worthwhile goal to pursue in future work.

Indeed, decreasing the latency of self-adaptive inference is an intriguing avenue for research. Using adaptive step sizes, higher-order optimization (Botev et al., 2017; Kingma & Ba, 2015), or implicit gradients (Rajeswaran et al., 2019), we may lower the number of required update steps. An orthogonal line of research may consider low-precision computations (Wang et al., 2019), low-rank decomposition (Jaderberg et al., 2014), or alternating update strategies (Tonioni et al., 2019) to offer reduced computational footprint for each update iteration. We are excited to explore these directions and hope the reader may find this work equally inspiring.

## Acknowledgments

This project is partially funded by the European Research Council (ERC) under the European Union's Horizon 2020 research and innovation programme (grant agreement No. 866008) as well as the State of Hesse (Germany) through the cluster project "The Adaptive Mind (TAM)".

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

# A  Baseline: Best practice implementation

To legitimately study the out-of-distribution model accuracy, it is essential to establish its upper bound attainable by the standard training procedures (Montavon et al., 2012). We followed the best practice in the literature and found a number of training details to be crucial for obtaining a highly competitive baseline, *i. e.*, a model that does not use our self-adaptive inference. Among them is using heavy data augmentation. Recall from Sec. 5 that we used random horizontal flipping, multi-scale cropping with a scale range of $[0.08, 1.0]$, as well as photometric image perturbations:* color jitter, random blur, and grayscaling. Color jitter, applied with probability 0.5, perturbs image brightness, contrast, and saturation using a factor sampled uniformly from the range $[0.7, 1.3]$. We use a different range of $[0.9, 1.1]$ for the hue factor. We randomly blur the image using a Gaussian kernel with the standard deviation sampled from $[0.1, 2.0]$. Additionally, we convert the image to grayscale with a probability of 0.1. Furthermore, we also found that the polynomial decay schedule we used for the learning rate, as well training for at least 50 epochs (for both GTA and SYNTHIA) are essential to achieve a high baseline accuracy. Note that we only used WildDash as the development set to tune these training details. We also experimented with higher input resolution and a larger batch size, but did not observe a significant improvement, yet a drastic increase in the computational overhead.

**On importance of the baseline.**  We note that the reported accuracy from previous work in Table 4 is not entirely consistent *w. r. t.* the choice of the model architecture. In particular, DRPC (Yue et al., 2019) uses an FCN, yet outperforms other domain generalization approaches with DeepLabv1 (Pan et al., 2018) and DeepLabv2 (Chen et al., 2020; 2021) considerably, as shown in Table 4. This is a regrettable consequence of inconsistent training schedules used in previous works that proved difficult to reproduce. For example, at the time of submission, the implementation by Yue *et al.* (Yue et al., 2019), which reports excellent segmentation accuracy for the baseline (*cf.* Table 4), has not yet been made available;† parts of the code implementing the semantic segmentation architecture introduced in (Pan et al., 2018) are also not publicly available.‡ These circumstances make reporting the accuracy of the implementation-specific baselines indispensable, which has thus become the standard practice in more recent previous (Chen et al., 2020; 2021) and related works (Gulrajani & Lopez-Paz, 2021).

# B  Self-adaptation: Additional analysis

## B.1  Analyzing self-adaptive normalization (SaN)

**Selecting $\alpha$.**  Fig. 5 shows a detailed plot of the influence of $\alpha$ on the segmentation accuracy, both on the development set of WildDash and on the target domains. We observe that the maximum accuracy on the development set is attained with $\alpha = 0.1$. Clearly, there is no guarantee that value 0.1 is the optimal one for the target domains. However, choosing $\alpha$ based on the development set is in line with the established practice in machine learning: Tuning model hyperparameters is not allowed on the test sets (*i. e.*, Cityscapes, BDD, IDD), but is only possible on the validation set (WildDash). In general, the hyperparameters found to be optimal on the validation set are not guaranteed to remain so on the test set, especially in the out-of-distribution scenario studied here. Nevertheless, self-adaptation is quite robust even to the inevitably suboptimal choice of the hyperparameters in the out-of-distribution setting. Despite $\alpha$ having been picked based on the validation dataset, our empirical results show consistent improvements over the baselines across all scenarios (*cf.* Table 3).

**SYNTHIA as the training set.**  Due to space constraints, we limited our study of SaN in the main paper to the scenario of using GTA as the source data (*cf.* Table 2). Here, we extend this study by training our models on SYNTHIA instead. Table 7 reports the segmentation accuracy (in terms of IoU) and the expected calibration error (ECE) for this case. Notably, SaN can still provide benefits for the expected segmentation accuracy if $p$-BN (*i. e.*, using target instance normalization statistics) fails to improve over the $t$-BN baseline, as is the case with ResNet-50 in Table 7; the results remain on par with the $t$-BN baseline even when $p$-BN is significantly worse than $t$-BN. In regard to calibration quality, the results are consistent with our model

---

*We use Pillow library (https://pillow.readthedocs.io) to implement photometric augmentation.

†https://github.com/xyyue/DRPC/issues

‡https://github.com/XingangPan/IBN-Net

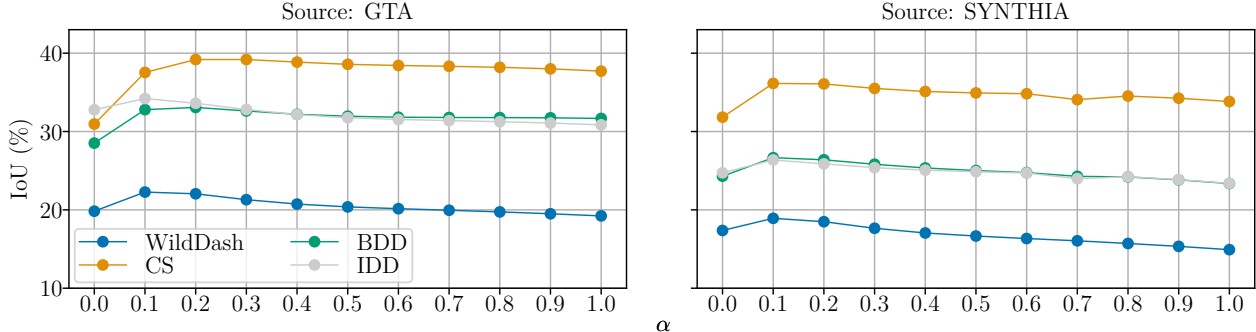

Figure 5: *Mean IoU (%, ↑) using SaN based on the optimal alpha on the development set (WildDash).* We report scores for the target domains (Cityscapes, BDD, IDD) for the ResNet-50 backbone after training on GTA *(left)* and SYNTHIA *(right).*

Table 6: *Mean IoU (%, ↑) comparison of SaN to alternatives: SN (Luo et al., 2019) and BIN (Nam & Kim, 2018).*

| Method | CS | BDD | IDD | Mean |
|---|---|---|---|---|
| SN | 31.75 | **33.60** | 31.60 | 32.32 |
| BIN | 34.57 | 32.68 | 30.22 | 32.49 |
| SaN *(ours)* | **37.54** | 32.79 | **34.21** | **34.85** |

trained on GTA (*cf.* Table 2(b)): Not only does SaN improve the prediction calibration of the baseline, it again exhibits a complementary effect with Monte Carlo dropout (Gal & Ghahramani, 2016). Overall, the combined results from Table 2 and Table 7 demonstrate that SaN improves both the model accuracy and the calibration quality of the predictions in the out-of-distribution setting irrespective of the backbone network and specifics of the source data.

**Comparison to other related work.** We additionally compare SaN to alternative normalization strategies proposed in the literature: Batch-Instance Normalization (BIN; Nam & Kim, 2018) and Switchable Normalization (SN; Luo et al., 2019). Although both of these techniques share some similarities with our SaN, these approaches were developed for different purposes. SN was only shown to improve in-domain accuracy, while BIN tackles domain adaptation for image classification, not domain generalization studied in this work. Furthermore, both methods modify the model architecture before training, while SaN works with any pretrained semantic segmentation model. Nevertheless, we implemented both SN and BIN in our segmentation model based on the ResNet-50 backbone. We trained these approaches on GTA in an identical setup as SaN. From results in Table 6, we observe that SaN outperforms both BIN and SN by a significant margin in terms of mean IoU of the target domains.

## B.2 Further design choices and runtime analysis

**Selecting parameters for self-adaptation.** In self-adaptation, we only update the model parameters in the layers `conv4_x`, `conv5_x`, and the classification head for the ResNet-50 backbone. Due to the higher computational cost of the ResNet-101 backbone, we only adapt the layers `conv5_x` and the classification head in this case. Note that we analyze the runtime costs for alternative configurations in Fig. 2 in the main text, as well as Fig. 6 below.

**Threshold and learning rate.** We investigate the influence of hyperparameters for self-adaptation: threshold $\psi$ and learning rate $\eta$. Table 9 reports the IoU after self-adaptation evaluated on the target domain Cityscapes after training with a ResNet-50 backbone on GTA. As already observed for $\alpha$ (*cf.* Fig. 5), self-adaptation is robust to the choice of hyperparameters.

Table 7: *(a) Segmentation accuracy using SaN.* We report the mean IoU(%, ↑) on three target domains (Cityscapes, BDD, IDD) across both backbones. In contrast to Table 2, we trained the networks on SYNTHIA in both cases. As before, *t*-BN denotes train BN (Ioffe & Szegedy, 2015), while *p*-BN refers to prediction-time BN (Nado et al., 2020). *(b) ECE for SaN and MC-Dropout (Gal & Ghahramani, 2016).* We report ECE scores (%, ↓) for three target domains (Cityscapes, BDD, IDD) across both backbones.

<table>
<tr><td colspan="5">(a)</td><td colspan="5">(b)</td></tr>
<tr><td rowspan="2">Method</td><td colspan="4">*IoU (%, ↑)*</td><td rowspan="2">Method</td><td colspan="4">*ECE (%, ↓)*</td></tr>
<tr><td>CS</td><td>BDD</td><td>IDD</td><td>Mean</td><td>CS</td><td>BDD</td><td>IDD</td><td>Mean</td></tr>
<tr><td>ResNet-50</td><td></td><td></td><td></td><td></td><td>ResNet-50</td><td>37.50</td><td>43.19</td><td>40.11</td><td>40.26</td></tr>
<tr><td>w/ *t*-BN</td><td>31.83</td><td>24.30</td><td>24.73</td><td>26.95</td><td>w/ SaN</td><td>30.96</td><td>33.27</td><td>36.31</td><td>33.51</td></tr>
<tr><td>w/ *p*-BN</td><td>33.83</td><td>23.36</td><td>23.39</td><td>26.86</td><td>w/ MC-Dropout</td><td>34.82</td><td>37.30</td><td>36.63</td><td>36.25</td></tr>
<tr><td>w/ SaN *(Ours)*</td><td>**36.14**</td><td>**26.66**</td><td>**26.37**</td><td>**29.72**</td><td>w/ both *(Ours)*</td><td>**30.66**</td><td>**33.06**</td><td>**35.60**</td><td>**33.11**</td></tr>
<tr><td>ResNet-101</td><td></td><td></td><td></td><td></td><td>ResNet-101</td><td>31.39</td><td>33.77</td><td>36.56</td><td>33.91</td></tr>
<tr><td>w/ *t*-BN</td><td>37.25</td><td>**29.32**</td><td>27.19</td><td>31.25</td><td>w/ SaN</td><td>30.33</td><td>31.83</td><td>36.26</td><td>32.81</td></tr>
<tr><td>w/ *p*-BN</td><td>34.58</td><td>24.24</td><td>22.32</td><td>27.05</td><td>w/ MC-Dropout</td><td>32.73</td><td>32.76</td><td>34.07</td><td>33.19</td></tr>
<tr><td>w/ SaN *(Ours)*</td><td>**38.01**</td><td>28.66</td><td>**27.28**</td><td>**31.32**</td><td>w/ both *(Ours)*</td><td>**27.71**</td><td>**30.48**</td><td>**32.67**</td><td>**30.29**</td></tr>
</table>

Table 8: *Mean IoU (%, ↑) for SaN and MC-Dropout (Gal & Ghahramani, 2016).* We report scores for three target domains (Cityscapes, BDD, IDD) on the ResNet-50 (He et al., 2016) backbone trained on GTA. We provided the ECE results in Table 2(b) to motivate our self-adaptive learning approach, which relies on improved model calibration. SaN also consistently improves the IoU of the MC-Dropout approach.

<table>
<tr><td rowspan="2">Method</td><td colspan="4">*IoU (%, ↑)*</td></tr>
<tr><td>CS</td><td>BDD</td><td>IDD</td><td>Mean</td></tr>
<tr><td>ResNet-50</td><td>30.95</td><td>28.52</td><td>32.78</td><td>30.75</td></tr>
<tr><td>w/ SaN *(Ours)*</td><td>37.54</td><td>32.79</td><td>34.21</td><td>34.85</td></tr>
<tr><td>w/ MC-Dropout</td><td>30.45</td><td>31.96</td><td>32.50</td><td>31.63</td></tr>
<tr><td>w/ both *(Ours)*</td><td>**38.84**</td><td>**35.13**</td><td>**35.55**</td><td>**36.50**</td></tr>
</table>

Table 9: *Mean IoU (%, ↑) for hyperparameter variation of threshold $\psi$ and learning rate $\eta$.* We report scores on the ResNet-50 backbone trained on GTA evaluated on Cityscapes for self-adaptation and SaN for reference.

<table>
<tr><td></td><td></td><td colspan="11">Self-Adaptation *(Ours): variation of* $\psi$ $(\eta = 0.05, N_t = 10)$</td></tr>
<tr><td></td><td>w/ SaN *(Ours)*</td><td>0.0</td><td>0.1</td><td>0.2</td><td>0.3</td><td>0.4</td><td>0.5</td><td>0.6</td><td>0.7</td><td>0.8</td><td>0.9</td><td>1.0</td></tr>
<tr><td>ResNet-50</td><td>37.54</td><td>44.00</td><td>44.00</td><td>44.01</td><td>44.04</td><td>44.24</td><td>44.54</td><td>44.91</td><td>**45.13**</td><td>44.59</td><td>43.62</td><td>39.40</td></tr>
<tr><td></td><td></td><td colspan="11">Self-Adaptation *(Ours): variation of* $\eta$ $(\psi = 0.7, N_t = 10)$</td></tr>
<tr><td></td><td>w/ SaN *(Ours)*</td><td>0.01</td><td>0.02</td><td>0.03</td><td>0.04</td><td>0.05</td><td>0.06</td><td>0.07</td><td>0.08</td><td>0.09</td><td>0.1</td><td></td></tr>
<tr><td>ResNet-50</td><td>37.54</td><td>44.40</td><td>44.77</td><td>44.95</td><td>45.06</td><td>**45.13**</td><td>45.08</td><td>45.06</td><td>45.07</td><td>45.04</td><td>44.96</td><td></td></tr>
</table>

**The choice of augmentation strategies.** We verify the influence of the augmentation type used by self-adaptation. Recall from Sec. 5.2 that we use multiple scales with horizontal flipping and grayscaling to augment one image sample. We compare a flipping-only, scaling-only, and grayscaling-only version of our self-adaptation to the combination of flipping, grayscaling, and the spatial scaling, which we used in the main text. We used a ResNet-50 backbone trained on GTA and report the accuracy on Cityscapes in Table 10. We observe a significant boost in accuracy in comparison to a strong baseline that uses our SaN with no augmentations. Furthermore, we show that using multiple scales is more important than flipping for self-adaptation. Note that the augmentations used do not impact the runtime in a significant way, since the batch sizes between these setups vary insignificantly; it is the backpropagation that dominates the main

Table 10: *The role of the augmentation type in self-adaptation.* We report mean IoU (%, ↑) and runtime (ms, ↓) for TTA (Simonyan & Zisserman, 2015) and our self-adaptation for the GTA source domain and the Cityscapes target domain for the ResNet-50 backbone.

| Method | TTA | | Ours | |
|---|---|---|---|---|
| | IoU | Runtime | IoU | Runtime |
| Baseline | 30.95 | **314** | 31.47 | **7135** |
| SaN | 37.54 | **314** | 39.04 | **7135** |
| Multiple scales | 42.27 | 749 | 44.92 | 7272 |
| Horizontal flipping | 38.01 | 986 | 39.33 | 7593 |
| Grayscaling | 37.96 | 908 | 39.65 | 7193 |
| Multiple scales + horizontal flipping | 42.48 | 1316 | 44.94 | 7890 |
| Multiple scales + grayscaling | 42.28 | 1202 | 45.06 | 7486 |
| Multiple scales + horizontal flipping + grayscaling | **42.56** | 1308 | **45.13** | 7862 |

Table 11: *Runtime with TTA (Simonyan & Zisserman, 2015) or self-adaptation.* We report the runtime (ms, ↓) for both ResNet-50 and ResNet-101 on three dominant resolutions of $2048 \times 1024$, $1280 \times 720$, and $1920 \times 1080$, corresponding to the target domains Cityscapes, BDD, and IDD, respectively.

| Method | Input resolution | | | |
|---|---|---|---|---|
| | CS | BDD | IDD | Mean |
| ResNet-50 (w/ SaN) | **314** | **136** | **214** | **221** |
|   TTA (w/ SaN) | 1308 | 713 | 766 | 929 |
|   Self-adaptation (*ours*) | 7862 | 3742 | 5307 | 5637 |
| ResNet-101 (w/ SaN) | **458** | **239** | **252** | **316** |
|   TTA (w/ SaN) | 1519 | 766 | 860 | 1048 |
|   Self-adaptation (*ours*) | 9060 | 4241 | 6142 | 6481 |

computational footprint. Varying the number of iterations, as studied in Sec. 5.1, provides a more flexible mechanism for accuracy-runtime trade-off.

**Inference time.** Table 11 compares the inference time of our self-adaptation *w. r. t.* test-time augmentation (TTA) and single-scale inference across a range of input resolutions available in the target domains. We obtain these results by running inference on a single NVIDIA GeForce RTX 2080 GPU. To improve the runtime estimate, for each dataset we average the inference time over the complete image set. This is to account for small deviations in the input resolution (*e. g.*, IDD mostly contains images of resolution $1920 \times 1080$, but also has images with resolution $1280 \times 720$). Since our self-adaptation uses 10 update iterations, the increase in the inference time *w. r. t.* TTA is expected. Although such cost may be detrimental for real-time applications, the significant accuracy benefits of self-adaptation (*cf.* Table 4) may find appeal in applications where the importance of the prediction quality outweighs the incurred overhead in the frame rate (*e. g.*, medical image analysis). Furthermore, self-adaption can amortize the runtime costs by using fewer update iterations, while still providing a clear accuracy improvement. For example, using $N_t = 3$ iterations decreases the runtime of self-adaptation almost twofold while preserving around 95% of the model accuracy attainable with more iterations.

We also investigated the influence of using the automatic mixed precision module in PyTorch and its influence on the inference runtime. While maintaining an identical IoU on Cityscapes using the ResNet-50 backbone, mixed precision with $N_t = 10$ reduced the runtime by almost 30%: the inference takes only 5746 ms compared to 7862 ms using single precision.

Table 12: *Mean IoU (%, ↑) using our self-adaptation integrated with DeepLabv3+ (Chen et al., 2018b) based on a ResNet-50 and ResNet-101 backbone as well as with HRNet-W18, HRNet-W48 (Wang et al., 2021b), and UPerNet (Xiao et al., 2018) with a Swin-T backbone (Liu et al., 2021).* We observe substantial improvements of the segmentation accuracy on all three target domains (Cityscapes, BDD, and IDD) after training on GTA.

| Method | Target domains | | | |
|---|---|---|---|---|
| | CS | BDD | IDD | Mean |
| DeepLabv3+ ResNet-50 (Baseline) | 37.51 | 35.45 | 37.50 | 36.82 |
|   Self-adaptation *(ours)* | **46.56** | **43.17** | **44.07** | **44.60** |
| DeepLabv3+ ResNet-101 (Baseline) | 38.19 | 37.05 | 38.22 | 37.82 |
|   Self-adaptation *(ours)* | **48.14** | **44.52** | **45.72** | **46.13** |
| HRNet-W18 (Baseline) | 33.08 | 29.40 | 32.97 | 31.82 |
|   Self-adaptation *(ours)* | **44.05** | **38.29** | **43.78** | **42.04** |
| HRNet-W48 (Baseline) | 34.66 | 30.85 | 34.64 | 33.38 |
|   Self-adaptation *(ours)* | **48.82** | **42.79** | **43.74** | **45.12** |
| UPerNet Swin-T (Baseline) | 39.67 | 36.04 | 39.74 | 38.48 |
|   Self-adaptation *(ours)* | **45.04** | **39.77** | **44.33** | **43.05** |

We additionally provide runtime-accuracy plots for GTA → BDD and GTA → IDD generalization in Fig. 6. The data supports our conclusions drawn on GTA → Cityscapes generalization (*cf.* Sec. 5.3) that *(i)* self-adaptation provides clear advantages in segmentation accuracy over baselines at a reasonable increase of the inference time; *(ii)* it is both more accurate and more efficient than model ensembles; and *(iii)* it exhibits a flexible runtime-accuracy trade-off by means of varying the number of update iterations and the number of the layers to adjust.

### B.3 Self-adaptation with state-of-the-art architectures

Our approach generalizes to more recent architectures. We trained five state-of-the-art segmentation models on GTA: DeepLabv3+ (Chen et al., 2018b) with both a ResNet-50 and ResNet-101 backbone, HRNet-W18 and HRNet-W48 (Wang et al., 2021b), as well as UPerNet (Xiao et al., 2018) with a Swin-T Transformer backbone (Liu et al., 2021). Table 12 reports consistent and substantial improvement of the mean IoU over the baseline, across all these architectures and the target domains.

### B.4 Evaluation on diverse domain shifts

The ACDC dataset (Sakaridis et al., 2021) offers densely labeled driving scenes under adverse weather conditions such as fog, rain, and snow. As far as we are aware, previous work on domain generalization for semantic segmentation did not consider this benchmark. Therefore, we are only able to compare our results to our own baseline.
We evaluate our Deeplabv1 model with a ResNet-50 backbone trained on GTA with and without our self-adaptation approach. Table 13 presents the results. Our method demonstrates a stark improvement across all weather conditions outperforming our baseline model by 13.57% on average. This experiment underscores strong multi-target domain generalization of self-adaptation, since we evaluate the same model as we did for other target datasets without any change (*cf.* Table 4).

### B.5 Qualitative examples

We provide additional qualitative results by running inference on three models: ResNet-101 trained on GTA in Fig. 7; ResNet-50 and ResNet-101 trained on SYNTHIA in Figs. 8 and 9, respectively. Similar to our observations in Sec. 5.3 (*cf.* main text), our approach exhibits more homogeneous semantic masks with visibly fewer jagged-shaped artifacts than the baseline (*e. g.*, "sidewalk" false positives in Fig. 7). Models with

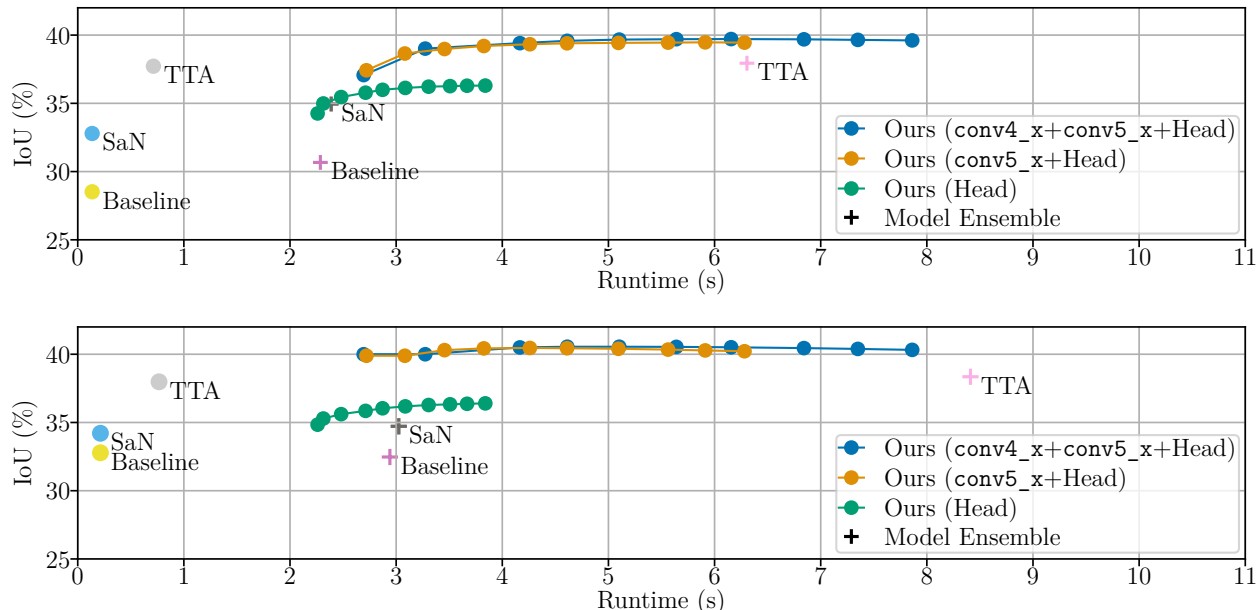

Figure 6: *Runtime-accuracy comparison on GTA → BDD (top) and GTA → IDD (bottom) generalization using ResNet-50.* The curves on this plot trace self-adaptation iterations, *i. e.* the first point corresponds to $N_t = 1$, while the last shows $N_t = 10$. While self-adaptation increases the inference time of the baseline and TTA for the sake of improved accuracy, it is still more efficient and accurate than model ensembles of 10 networks. The choice of the layers for self-adaptation updates (the naming follows He et al., 2016) further provides a favorable runtime-accuracy trade-off. The runtime is computed on a single NVIDIA GeForce RTX 2080 GPU.

Table 13: *Mean IoU (%, ↑) using our self-adaptation integrated with DeepLabv1 based on a ResNet-50 backbone.* We observe substantial improvements of the segmentation accuracy under adverse weather conditions (fog, rain, and snow) on the ACDC (Sakaridis et al., 2021) validation set after training on the synthetic GTA.

| Method | *Weather conditions* | | | |
|---|---|---|---|---|
| | Fog | Rain | Snow | Mean |
| ResNet-50 (Baseline) | 26.12 | 27.32 | 24.64 | 26.03 |
| Self-adaptation *(ours)* | **41.61** | **37.54** | **39.66** | **39.60** |

self-adaptation may still struggle with cases of mislabeling regions with an incorrect, but semantically related class. For example, the model often assigns "sidewalk" to the road pixels from BDD and IDD in Figs. 8 and 9. This is an expected outcome if the erroneous labels are already contained in the pseudo-labels of the initial prediction, on which self-adaptation relies. These failure cases occur more frequently if the domain shift between the train and the test distributions is more significant, such as between SYNTHIA and BDD, which can lead to poorly calibrated predictions. Since applying SaN alone results in improved calibration (*cf.* Table 2(b)), it alleviates this issue, and hence self-adaptation can cope well with milder domain shift scenarios, as a result. As the examples on Cityscapes and Mapillary in Figs. 8 and 9 show, despite the baseline model exhibiting some degree of this failure mode, our inference method visibly rectifies these errors.

We additionally ran our inference on video sequences and include the results as part of our supplementary material. Confirming our previous analysis of the qualitative results (*cf.* Sec. 5.2), we observe that our approach clearly improves the segmentation quality and removes some of the most pathological failure modes of the baseline (*e. g.,* the lower middle part of the frame area).

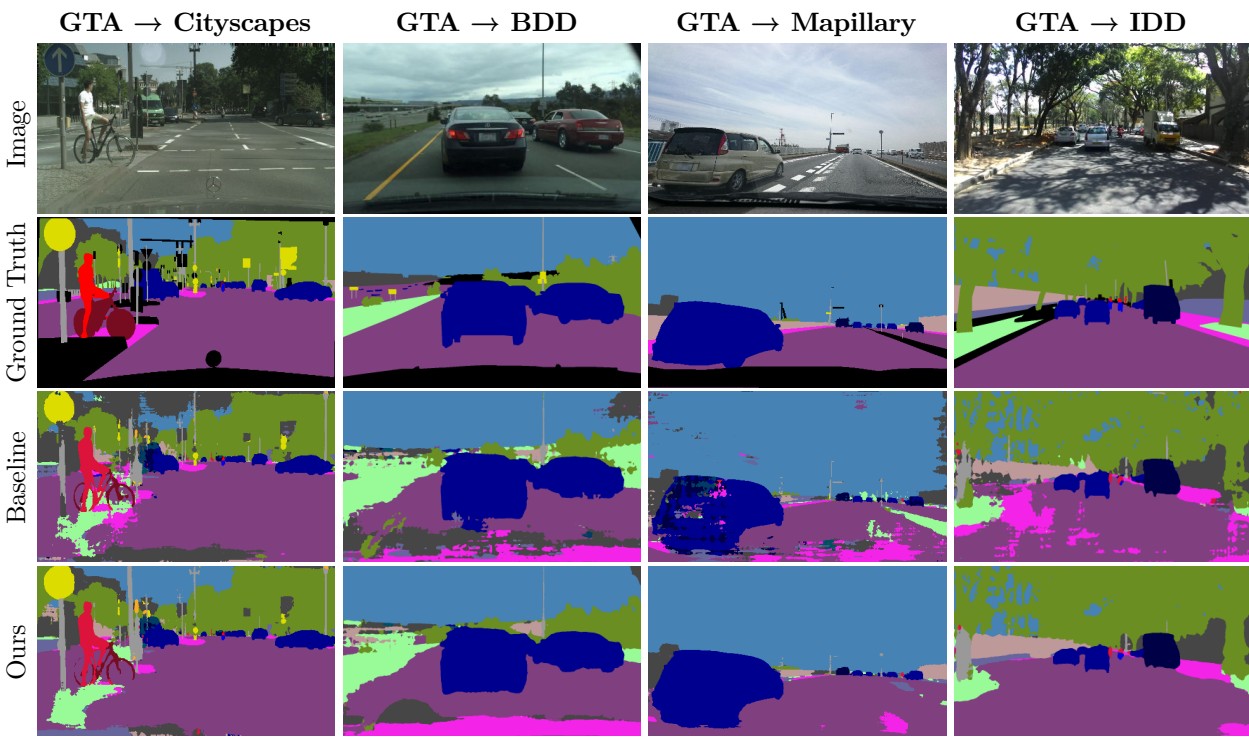

Figure 7: *Qualitative semantic segmentation results for generalization from GTA to Cityscapes, BDD, Mapillary, and IDD* for the ResNet-101 backbone.

### B.6 Failure Cases

Conceptually, if the initial semantic prediction is incorrect and confident, which may happen due to imperfect model calibration, such error is likely to end up in the pseudo-labels and lead astray the self-adaptation process. The rightmost columns in Figs. 8 and 9 already visualized this failure mode: the baseline assigns large areas of the "road" class to the "sidewalk" category. We have further extended these examples with Fig. 10. Misguided by incorrect pseudo-labels, our self-adaptation may exacerbate this issue, *i. e.*, propagate the incorrect label to the areas sharing the same appearance. To gain further insights, we investigated this issue statistically. The histograms in Fig. 11 show relative improvement of our self-adaptive inference strategy *w. r. t.* the baseline in terms of IoU on four validation sets. We conclude that such decrease in segmentation accuracy is actually quite rare: less than 10% of the images, on average, exhibit lower accuracy. In most such cases the accuracy reduces only marginally (by less than 5%), and only a fraction of images – less than 1% on average – deteriorate in accuracy by at most 10%. The overwhelming majority of the image samples benefit from our self-adaptive process, which increases their accuracy by up to 35% IoU compared to the baseline.

## C Dataset details

**GTA.** GTA (Richter et al., 2016) is a street view dataset generated semi-automatically from the computer game Grand Theft Auto V. The dataset consists of 12,403 training images, 6,382 validation images, and 6,181 testing images of resolution $1914 \times 1052$ with 19 different semantic classes.

**SYNTHIA.** We use the SYNTHIA-RAND-CITYSCAPES subset of the synthetic dataset SYNTHIA (Ros et al., 2016), which contains 9,400 images, and has 16 semantic classes in common with GTA. Images have a resolution of $1280 \times 760$ pixels.

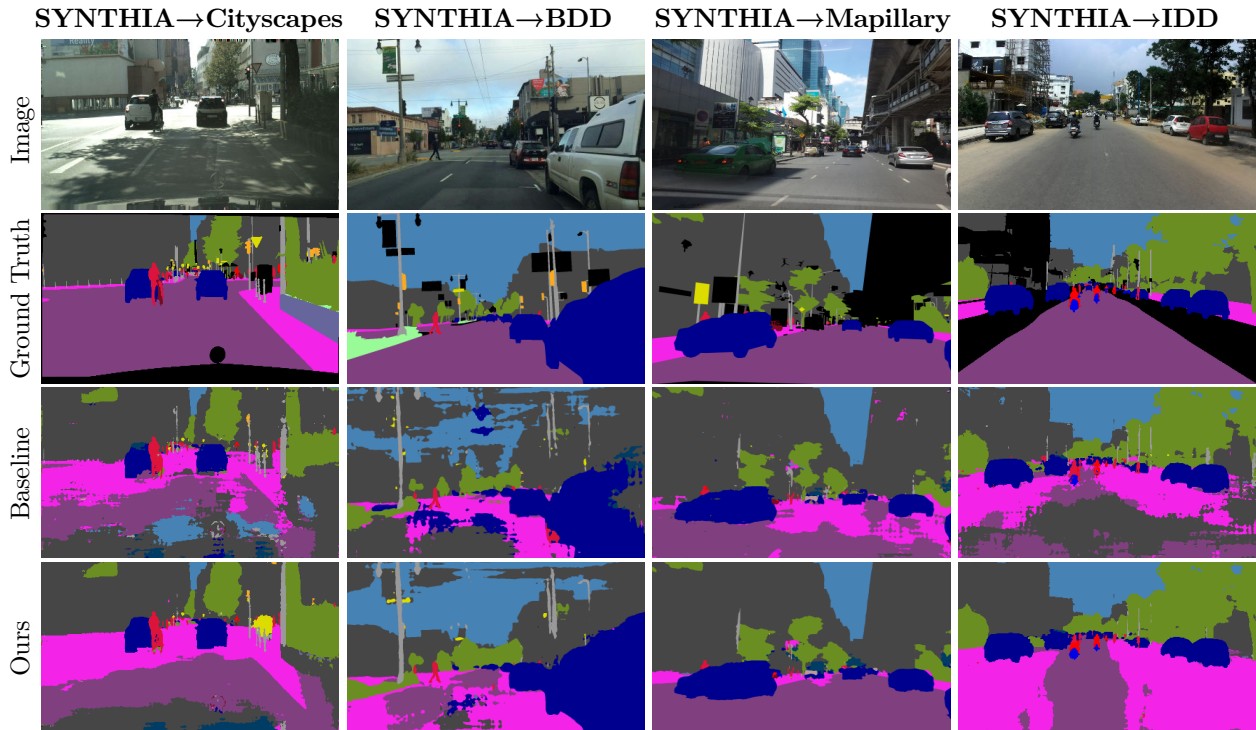

Figure 8: *Qualitative semantic segmentation results for generalization from SYNTHIA to Cityscapes, BDD, Mapillary, and IDD* for the ResNet-50 backbone.

**WildDash.** The WildDash benchmark (Zendel et al., 2018) was developed to evaluate models regarding their robustness for driving scenarios under real-world conditions. It comprises 4,256 images of real-world scenes with a resolution of $1920 \times 1080$ pixels.

**Cityscapes.** Cityscapes (Cordts et al., 2016) is an ego-centric street-scene dataset and contains 5,000 high-resolution images with $2048 \times 1024$ pixels. It is split into 2,975 train, 500 val, and 1,525 test images with 19 semantic classes being annotated.

**BDD.** BDD (Yu et al., 2020) is a driving video dataset, which also contains semantic labelings with the identical 19 classes as in the other datasets. Images have a resolution of $1280 \times 720$ pixels. The training, validation, and test sets contain 7,000, 1,000, and 2,000 images, respectively.

**IDD.** IDD (Varma et al., 2019) is a dataset for road scene understanding in unstructured environments. It contains 10,003 images annotated with 34 classes even though we only evaluate on the 19 classes overlapping with the other datasets. IDD is split into 6,993 training images, 981 validation images, and 2,029 test images.

**Mapillary.** Annotations from Mapillary (Neuhold et al., 2017) contain 66 object classes; analogously to IDD we only evaluate on the 19 classes overlapping with the other datasets. The dataset is split into a training set with 18,000 images and a validation set with 2,000 images with a minimum resolution of $1920 \times 1080$ pixels.

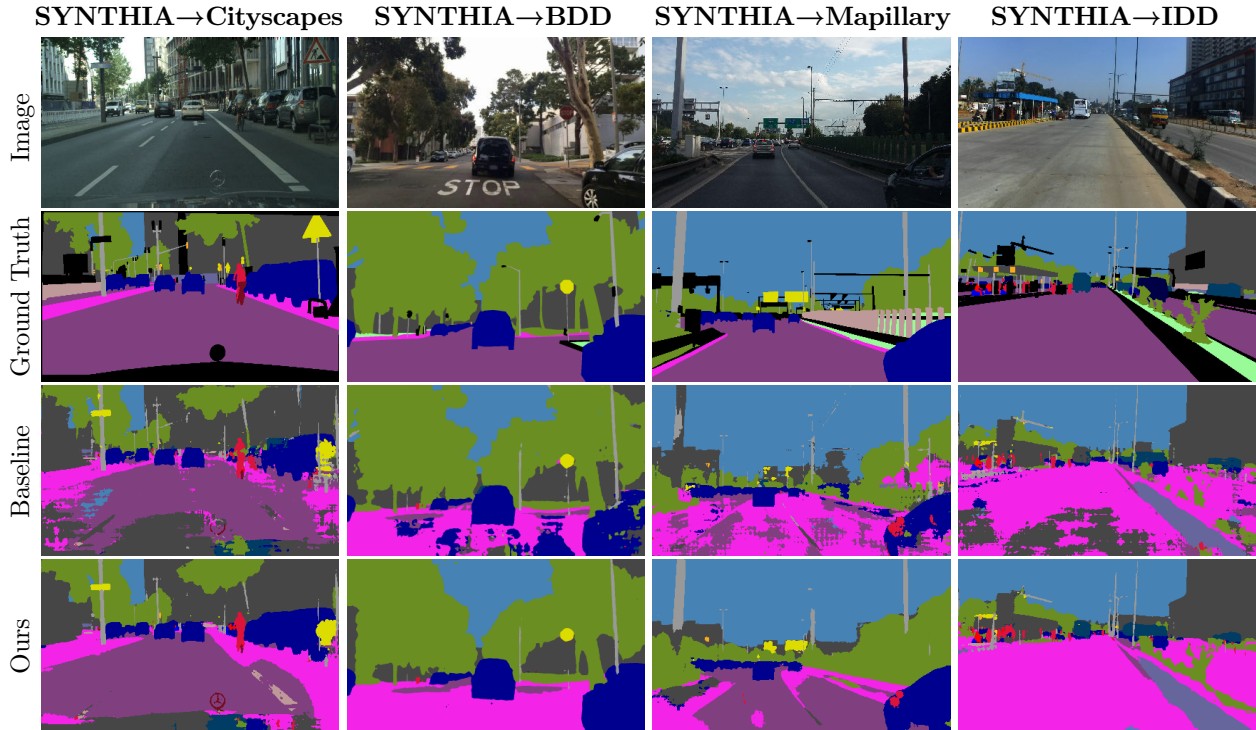

Figure 9: *Qualitative semantic segmentation results for generalization from SYNTHIA to Cityscapes, BDD, Mapillary, and IDD* for the ResNet-101 backbone.

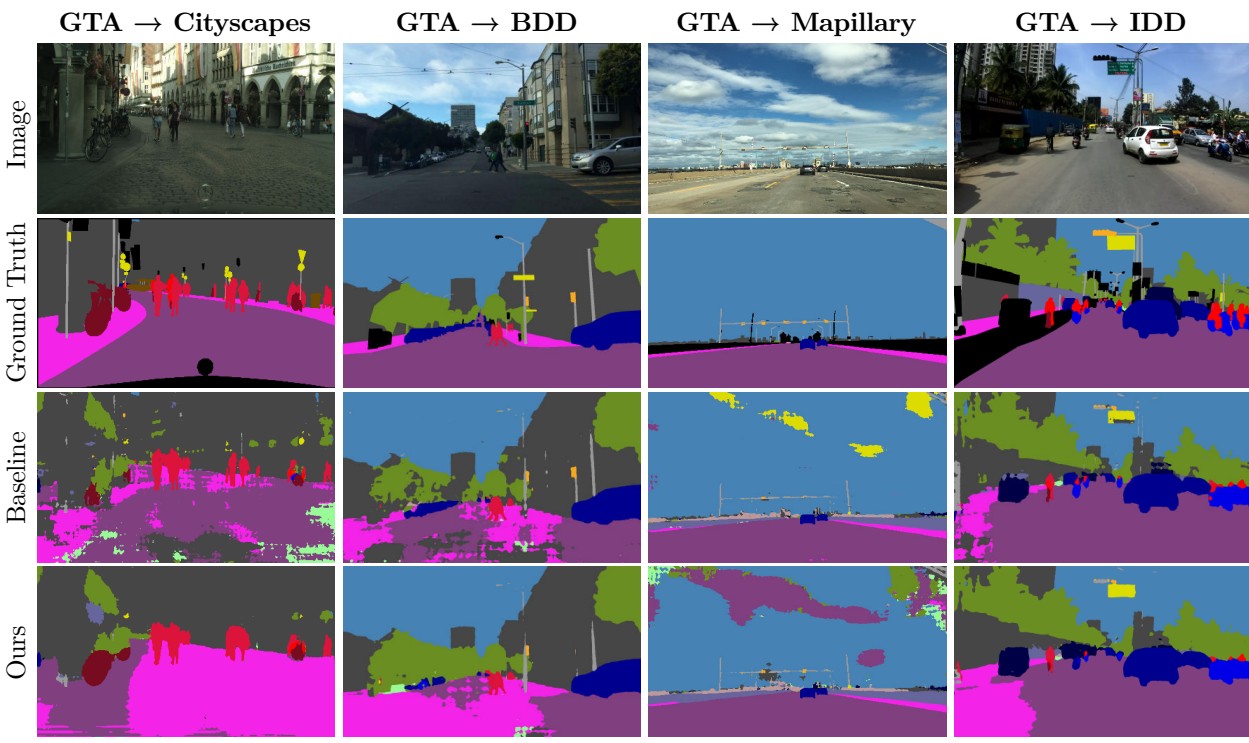

Figure 10: *Failure cases of semantic segmentation for generalization results from GTA to Cityscapes, BDD, Mapillary, and IDD* for the ResNet-50 backbone.

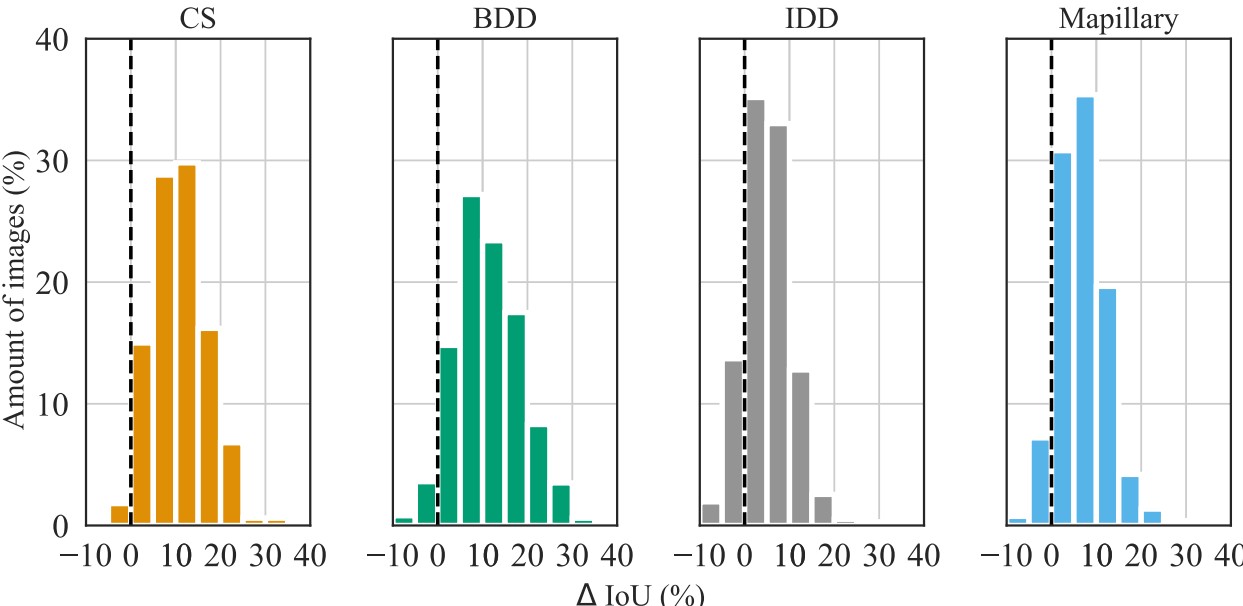

Figure 11: *Empirical distribution of the IoU change of individual images for generalization from source domain (GTA) to target domains (Cityscapes, BDD, IDD and Mapillary)* for the ResNet-50 backbone. We visualize the relative improvement of our self-adaptive inference strategy $w.r.t.$ the baseline in terms of accuracy with $\Delta$ IoU (%).

