# OpenReview forum: "Semantic Self-adaptation: Enhancing Generalization with a Single Sample"
_TMLR — Accepted by TMLR_

### Review · Reviewer_TqUP · 2023-03-25

**Summary Of Contributions:**

Deep networks for semantic segmentation lack out-of-domain generalization. In this work, a self-adaptive approach is proposed to adjust the inference process for each input sample, fine-tuning parameters, and interpolating in Batch Normalization layers. This approach sets a new state-of-the-art accuracy on generalization benchmarks, complementing model regularization for improving generalization to out-of-domain data.

**Audience:**

Yes

**Broader Impact Concerns:**

I think in general this work is interesting to the community of domain adaptation, domain transfer, and semantic segmentation.
Test-time adaptation is definitely useful for solving domain gaps and perhaps synthetic-to-real generalization. However, computation and time costs could limit the possible use cases of this work. I hope authors could for example enumerate useful use cases, to verify the true broad impact of this work in the real world.

**Claims And Evidence:**

Yes

**Requested Changes:**

Introducing test-time adaptation (both parameter label and BN level) involves more hyperparameters to tune, like $\psi$, augmentations, $N_t, \eta$. In more diverse domain gaps (low resolution, weather changes, low light, etc.), the proposed method may not guarantee that it is robust to the tuning of these hyperparameters. I hope the authors could justify if the proposed method is user-friendly or any ablation study on the sensitivity of the method to these hyperparameters.

**Strengths And Weaknesses:**

Strength
* Paper is well written.
* Authors provide comprehensive experiments.

Weakness:
* Under which scenario(s) would we like to pay this computation/time cost for single image adaptation? Examples in Figure 1 are from driving scenes. I suppose driving scene understanding is highly latency-sensitive.
* Furthermore, making predictions on a batch of samples can speed up the inference time. Will this method be adopted for multi-sample adaptation during inference? If so, will any study about the impact of inference batch size on mIoU be interesting?
* As discussed in the abstract, two core parts of the method is previously proposed. I hope authors could better explain/claim their core novelty.

---

> ### Author Response · Authors · 2023-04-25
> **Response to Reviewer TqUP [1/2]**
>
> Thank you for your valuable feedback.
>
> > **Q1: Application scenarios**
>
> We acknowledge that self-adaptation in the proposed form is not real-time, while our empirical analysis centers on traffic scenes, where low latency is desired. However, our intent of using this evaluation scenario is to compare self-adaptation to a large body of previous work, as this has been the established testbed for domain generalization in the context of semantic segmentation.
> There are numerous applications where accuracy of a semantic segmentation model is more important than its efficiency. For example, in (e.g., outdoor) 3D scene reconstruction and SLAM systems, of which many are not real-time, it is important to identify static parts of the scene. This information can be provided by a semantic segmentation model (e.g., using categories “road”, “building”, etc.). Another example is scene editing for the purpose of data protection. One may want to identify (and erase) faces of pedestrians, license plates on vehicles as data preparation. Semantic segmentation would help identify such areas of interest.
> We remark that in contrast to some previous work (e.g. TENT), we highlight and thoroughly analyze the increased computational expense for transparency. We hope this analysis inspires follow-up efforts toward improved efficiency.
>
> > **Q2: Multi-sample adaptation**
>
> We expect that using multiple samples would amortize the runtime of self-adaptation and improve its accuracy. Indeed, this scenario was thoroughly analyzed by Nado et al. [1] for image recognition. The issue with employing the same strategy here is that it would disadvantage prior art, which runs inference on one image at a time, hence invalidating the comparison. We adhere to the domain generalization scenario, where no knowledge between data samples must be shared.
>
> > **Q3: Novelty**
>
> Our core novelty lies in providing a fresh perspective on domain generalization for semantic segmentation: previous work modified the training process or model architectures, whereas we adjust only the inference process following conventional training. Our extensive empirical analysis shows that it is surprisingly effective and outperforms previous works from ca. four years of research.
>
>
> > **Q4: Hyperparameter sensitivity**
>
>
> As it holds for any machine learning model, hyperparameters chosen on the validation set are not guaranteed to be optimal on the test set. Indeed, we are fully transparent in not using the test domains for hyperparameter tuning – in contrast to previous work. Note that
> * We use the same set of hyperparameter values for all test domains.
> * The target datasets (e.g. BDD) already exhibit a large diversity of image resolution, weather and low-light conditions. Furthermore, we extend the diversity of the standard domain generalization benchmark by another dataset (IDD) in our evaluation. Nevertheless, we additionally evaluated self-adaptation on another dataset ACDC (see below and Sec. B.4).
> * Self-adaptation is robust to the choice of hyperparameters, as we show empirically next.
>
>
> In the following experiments, we report mIoU on Cityscapes after self-adaptation of a model trained on GTA.
>
> Variation of threshold $\psi$:
>
> | $\psi$|SaN Baseline |     0.0     |     0.1    |      0.2    | 0.3    | 0.4    | 0.5    | 0.6    | 0.7    | 0.8    | 0.9    | 1.0    |
> |--------  |:--------:|:--------:|:--------:|:--------:|:--------:|:--------:|:--------:|:--------:|:--------:|:--------:|:--------:|:--------:|
>  | |37.54 |44.00| 44.00| 44.01| 44.04| 44.24| 44.54| 44.91| 45.13| 44.59| 43.62| 39.40|
>
> Note that using $\psi = 1.0$ means filtering out every prediction, hence not using any pseudo label. We include this number for the sake of completeness.
>
> Learning rate $\eta$:
>
> | $\eta$|    SaN Baseline | 0.01     |     0.02    |0.03     |0.04     |0.05     |0.06    |0.07     |0.08     |0.09     |0.1     |
> |-------- |:--------: |:--------:|:--------:|:--------:|:--------:|:--------:|:--------:|:--------:|:--------:|:--------:|:--------:|
>  ||37.54 | 44.40| 44.77| 44.95| 45.06| 45.13| 45.08| 45.06| 45.07| 45.04| 44.96|
>
> We observe that self-adaptation performs robustly across substantial variation of the learning rate.
>
> We have added these experiments to the revision of the paper in Sec. B.2. Please refer to Fig. 2 for the effect of $N_t$. It shows that already few iterations (e.g. 3) significantly improve the baseline accuracy.

---

> ### Author Response · Authors · 2023-04-25
> **Response to Reviewer TqUP [2/2]**
>
> > **Q5: More diverse domain gaps**
>
> We are happy to explore more diverse domain gaps. The ACDC dataset [2] offers densely labeled driving scenes under challenging visual conditions such as fog, rain, and snow. We evaluate our ResNet-50 model trained on GTA with and without our self-adaptation approach. Our method demonstrates a remarkable and consistent improvement across all weather conditions. This experiment further underscores the robustness of our hyperparameter selection. Note that, to the best of our knowledge, other domain generalization methods do not evaluate on this benchmark, hence we only show a comparison to our baseline. We have added the results to Sec. B.4 in the revision of our paper.
>
> | Method            | Fog    | Rain   | Snow   | Avg    |
> |-------------------|--------|--------|--------|--------|
> | ResNet50 (Baseline) | 26.12 | 27.32 | 24.64 | 26.03 |
> | w/ Self-adaptation (Ours)           | 41.61 | 37.54 | 39.66 | 39.60 |
>
>
> > **Q6: Broad impact**
>
> We are happy to include a Broader Impact Statement in our manuscript if there are any concerns on the ethical implications of our work.
>
> [1] Nado et al., Evaluating Prediction-Time Batch Normalization for Robustness under Covariate Shift, arXiv 2020\
> [2] Sakaridis et al., ACDC: The Adverse Conditions Dataset with Correspondences for Semantic Driving Scene Understanding, ICCV 2021

---

### Review · Reviewer_eXNC · 2023-04-15

**Summary Of Contributions:**


In this study, the authors study the generalization in semantic segmentation with synthetic data using adaptation techniques. They propose a self-adaptive approach that adjusts the inference procedure for each individual test sample. This approach involves creating a mini-batch from the given test sample using data augmentation techniques, averaging the softmax probabilities, and generating a pseudo label based on a class-dependent confidence threshold. The model parameters are then updated by minimizing the cross-entropy loss with respect to the pseudo label. This process is repeated for a few iterations before producing the final prediction and discarding the updated model after prediction.
The authors also highlight the lack of systematic evaluation in existing studies and propose steps for a rigorous and systematic evaluation of domain generalization methods for semantic segmentation. Their proposal includes using multiple domains in the test set, employing a single model for all test points, independently selecting the model regardless of the test samples, and clearly specifying the validation data.
Following the revised evaluation protocol, the authors conduct a comprehensive empirical evaluation by comparing their proposed method with several state-of-the-art baselines on multiple datasets and models. They observe that their proposed method outperforms the baselines.


**Audience:**

Yes

**Claims And Evidence:**

Yes

**Requested Changes:**

1. Could you please summarize the method in an algorithm block?

2. Could you please provide reasoning behind the method’s effectiveness. It could be either via some controlled synthetic experiments or even better with some theoretical analysis.


**Strengths And Weaknesses:**


Strengths:
----------------
1. Novel approach: The authors propose a self-adaptive approach for semantic segmentation that adjusts the inference procedure for each sample, which is a unique and innovative approach to address the generalization problem from synthetic data. This can potentially lead to improved performance in semantic segmentation tasks.

2. Rigorous evaluation protocol: The authors highlight the lack of systematic evaluation in existing works and propose a comprehensive evaluation protocol that includes the use of multiple domains in the test set, employing a single model for all test points, independently selecting the model regardless of the test samples, and clearly specifying the validation data. This rigorous evaluation protocol enhances the reliability of the research findings.

3. Empirical evaluation: The authors conduct a thorough empirical evaluation by comparing their proposed method with several state-of-the-art baselines on multiple datasets and models. This extensive evaluation provides evidence of the effectiveness of their approach and adds credibility to their findings.


Weaknesses
---------------
1. Although the idea to update model during inference is interesting, it is going to be make the inference procedure computationally expensive. Usually inference is expected to be very fast.

2. Lack of reasoning or rationale behind the method. In particular why should the proposed method work well?

---

> ### Author Response · Authors · 2023-04-25
> **Response to Reviewer eXNC**
>
> Thank you for your valuable feedback.
>
> > **Q1: Runtime**
>
> We acknowledge that self-adaptation in the proposed form is not real-time, while our empirical analysis centers on traffic scenes, where low latency is desired. However, our intent of using this evaluation scenario is to compare self-adaptation to a large body of previous work, as this has been the established testbed for domain generalization in the context of semantic segmentation.
> There are numerous applications where accuracy of a semantic segmentation model is more important than its efficiency. For example, in (e.g., outdoor) 3D scene reconstruction and SLAM systems, of which many are not real-time, it is important to identify static parts of the scene. This information can be provided by a semantic segmentation model (e.g., using categories “road”, “building”, etc.). Another example is scene editing for the purpose of data protection. One may want to identify (and erase) faces of pedestrians, license plates on vehicles as data preparation. Semantic segmentation would help identify such areas of interest.
> We remark that in contrast to some previous work (e.g. TENT), we highlight and thoroughly analyze the increased computational expense for transparency. We hope this analysis inspires follow-up efforts toward improved efficiency.
>
> > **Q2: Reasoning behind the method**
>
> Some insight about the self-adaptive normalization can be derived from previous studies, e.g., Nado et al. [1] (p. 5, “Prediction-Time Batch Normalization Repairs Mismatched Supports”) and Schneider et al. [2]. The idea is that due to the distribution shift, the mean and the standard deviation estimated from the synthetic data is not representative of the test distribution. The interpolation strategy that we used is a modality-preserving approximation of the test distribution. It is intuitive, simple to implement and works well in practice. We would be happy to experiment with alternative strategies if advised by reviewers.
>
> Training on pseudo-labels, as we do in self-adaptation, has been a successful and practical strategy in semi-supervised learning and unsupervised domain adaptation. We agree that much remains to be understood theoretically, and previous studies have been limited to simple models so far (e.g., Zhang et al. [3]). Conducting a similar theoretical analysis is beyond the scope of the present work.
>
> The rationale behind combining both existing techniques becomes clear once we consider the improvement in model calibration (cf. Tab. 2b). Since the prediction confidence of a calibrated model is equivalent to the expected accuracy, this results in a low error rate of the pseudo-labels. For example, with a threshold $\psi = 0.7$, around 70% of the pseudo-labels produced by a well-calibrated model will be correct.
>
> > **Q3: Algorithm Block**
>
> Thanks for this suggestion. We have included an algorithm block as part of Sec. B.7.
> In summary, self-adaptation works as follows:
>
> 1. Train segmentation model on synthetic data (best-practice, established methodology).
> 2. Replace BN with SaN.
> 3. Tune hyperparameter $\alpha$ in SaN on the validation set (WildDash).\
> — Inference. Initial model parameters: $\theta_0$ —
> 4. For each test sample:\
> a. Obtain $\theta^\ast$ by minimizing the cross-entropy loss w.r.t. pseudo-labels (see Eq. 4).\
> b. Obtain segmentation masks using model parameters $\theta^\ast$.\
> c. Reset model parameters to $\theta_0$.
>
> [1] Nado et al., Evaluating Prediction-Time Batch Normalization for Robustness under Covariate Shift, arXiv 2020\
> [2] Schneider et al., Improving robustness against common corruptions by covariate shift adaptation, NeurIPS 2020\
> [3] Zhang et al., How unlabeled data improve generalization in self-training? A one-hidden-layer theoretical analysis, ICLR 2022

---

### Review · Reviewer_ycjh · 2023-04-16

**Summary Of Contributions:**

This paper presents an approach for domain-adaptive semantic segmentation. The basic assumption behind this setting is that the distribution and the training and validation data are not the same, which is of practical importance. The authors propose a self-adaptive algorithm that consists of two major components. First, it fine-tunes some convolutional layers using the test inputs with consistency regularization. Second, it interpolates between the source and the target statistics in Batch Normalization layers. Experimental results validate the effectiveness of this approach.

**Audience:**

Yes

**Broader Impact Concerns:**

I have no concerns about the broader impacts and the ethical implications.

**Claims And Evidence:**

Yes

**Requested Changes:**

See the Weaknesses above. I think the authors should clarify their claims on  'Enhancing Generalization with a Single Sample' .

**Strengths And Weaknesses:**

Strengths:

- This paper is well-organized. The writing is of high quality.
- This problem is important, and the proposed method shows strong empirical performance.
- The experimental results are extensive.

Weaknesses:

- In my opinion, the claim 'Enhancing Generalization with a Single Sample' is not sufficiently convincing. The proposed method introduces many tunable hyper-parameters. For example, in the fine-tuning process and adjusting the statistics of BN layers. The values of these hyper-parameters need to be determined in the validation set. Hence, more exactly, more than a single sample is leveraged in this process.

---

> ### Author Response · Authors · 2023-04-25
> **Response to Reviewer ycjh**
>
> Thank you for your valuable feedback.
>
> > **Q1: Single Sample Adaptation**
>
> Self-adaptation exhibits a high degree of tolerance to the change of the hyperparameter values.
> Therefore, it can improve the baseline accuracy even without the validation set, e.g., if we were to select the hyperparameters heuristically. This is supported by:
> - Fig. 4, which shows that $\alpha$ in self-adaptive normalization (SaN)  can be selected from a wide range in (0, 0.5]
> - Our following experiments on sensitivity analysis
>
> In the following experiments, we report mIoU on Cityscapes after self-adaptation of a model trained on GTA.
>
> Variation of threshold $\psi$:
>
> | $\psi$|SaN Baseline |     0.0     |     0.1    |      0.2    | 0.3    | 0.4    | 0.5    | 0.6    | 0.7    | 0.8    | 0.9    | 1.0    |
> |--------  |:--------:|:--------:|:--------:|:--------:|:--------:|:--------:|:--------:|:--------:|:--------:|:--------:|:--------:|:--------:|
>  | |37.54 |44.00| 44.00| 44.01| 44.04| 44.24| 44.54| 44.91| 45.13| 44.59| 43.62| 39.40|
>
> Note that using $\psi = 1.0$ means filtering out every prediction, hence not using any pseudo label. We include this number for the sake of completeness.
>
> Learning rate $\eta$:
>
> | $\eta$|    SaN Baseline | 0.01     |     0.02    |0.03     |0.04     |0.05     |0.06    |0.07     |0.08     |0.09     |0.1     |
> |-------- |:--------: |:--------:|:--------:|:--------:|:--------:|:--------:|:--------:|:--------:|:--------:|:--------:|:--------:|
>  ||37.54 | 44.40| 44.77| 44.95| 45.06| 45.13| 45.08| 45.06| 45.07| 45.04| 44.96|
>
> We observe that self-adaptation performs robustly across substantial variation of the learning rate.
>
> We have added these experiments to the revision of the paper in Sec. B.2. Please refer to Fig. 2 for the effect of $N_t$. It shows that already few iterations (e.g., 3) significantly improve the baseline accuracy.
>
> Further note that:
> - The number of hyperparameters in self-adaptation is four, which is comparable to previous work e.g.:
>     - DRPC [1] – 6 hyperparameters (K: number of ImageNet categories, $\lambda_1$ - $\lambda_5$: weights pyramid consistency loss)
>     - PIN [2] - 5 hyperparameters ($\lambda_1$ $\lambda_2$ weights loss, $m$ momentum memory network, $\alpha$ learning rate meta training step, $\beta$ learning rate meta testing step)
> - Some previous work (e.g., DRPC [1], WildNet [3]) uses non-synthetic data (e.g., ImageNet) already at training time.
> - Some previous work uses test data for hyperparameter tuning (e.g., FSDR [4], https://github.com/jxhuang0508/FSDR/issues/2#issuecomment-910089417), while most other works (e.g., RobustNet [5]) do not provide details on the validation process at all.
>
> [1] Yue et al., “Domain Randomization and Pyramid Consistency: Simulation-to-Real Generalization without Accessing Target Domain Data”, ICCV 2019\
> [2] Kim et al., “Pin the Memory: Learning to Generalize Semantic Segmentation”, CVPR 2022\
> [3] Lee et al., “WildNet: Learning Domain Generalized Semantic Segmentation from the Wild”, CVPR 2022\
> [4] Huang et al., “FSDR: Frequency Space Domain Randomization for Domain Generalization”, CVPR 2021\
> [5] Choi et al., “RobustNet: Improving Domain Generalization in Urban-Scene Segmentation via Instance Selective Whitening”, CVPR 2021

---

### Decision · Action_Editors · 2023-06-15

**Recommendation:** Accept with minor revision

**Comment:**

This paper proposes a self-adaptive semantic segmentation approach that enhances the generalization capability in the presence of domain shifts. The idea is both simple and effective, the paper is well-written, and the experimental verification is solid. Therefore, the AE recommends Accept with minor revision.

The authors are asked to make two vital revisions to the paper organization: 1) Move the Algorithm Block (i.e., B.7 and Algorithm 1 in the Appendix) to the main body of the paper. 2) Provide a detailed analysis of the impact of hyperparameters and a formal guideline for tuning them in the main body of the paper.


**Audience:**

Yes.

**Claims And Evidence:**

Yes.

---

> ### Author Response · Authors · 2023-07-18
> **Revision**
>
> Thank you very much for the favorable decision and the valuable feedback!
> We have incorporated your suggestions and the additional material from the rebuttal into the manuscript.
>
> - The algorithm block now appears in Section 3 (reviewer eXNC).
> - Section 5.4 now features the analysis of hyperparameter sensitivity with cross-references to related discussions (Reviewers ycjh and TqUP).
> - Appendix B.4 now includes experimental results on ACDC, extending the set of target domains from the main text (Reviewer TqUP).